

# Antarctic climate response in Last-Interglacial simulations using the Community Earth System Model (CESM2)

Mira Berdahl[1,*], Gunter R. Leguy[2,*], William H. Lipscomb[2], Bette L. Otto-Bliesner[2], Esther C. Brady[2], Robert A. Tomas[2], Nathan M. Urban[3], Ian Miller[4], Harriet Morgan[5], and Eric J. Steig[1]

[1]Department of Earth and Space Sciences, University of Washington, Seattle, WA, USA
[2]Climate and Global Dynamics Laboratory, NSF National Center for Atmospheric Research, Boulder, CO, USA
[3]Computational Science Initiative, Brookhaven National Laboratory, Upton, NY, USA
[4]Washington Sea Grant, Port Angeles, WA, USA
[5]Washington Department of Fish and Wildlife, Olympia, WA, USA
[*]These authors contributed equally to this work

**Correspondence:** Mira Berdahl (mberdahl@uw.edu), Gunter Leguy (leguy@ucar.edu)

**Abstract.**

We examine results from two transient modelling experiments that simulate the Last Interglacial period (LIG) using the state-of-the-art Community Earth System Model (CESM2), with a focus on climate and ocean changes relevant to the possible collapse of the Antarctic ice sheet. The experiments simulate the early millennia of the LIG warm period using orbital forcing, greenhouse gas concentrations and vegetation appropriate for 127 ka; in the first case (*127ka*) no other changes are made; in the second case (*127kaFW*), we include a 0.2 Sv freshwater forcing in the North Atlantic. Both are compared with a pre-industrial control simulation (*piControl*). In the *127ka* simulation, the global average temperature is only marginally warmer (0.004°C) than in the *piControl*. When freshwater forcing is added (*127kaFW*), there is surface cooling in the NH and warming in the SH, consistent with the bipolar seesaw effect. Near the Antarctic ice sheet, the *127ka* simulation generates notable ocean warming (up to 0.4°C) at depths below 200 m compared to the *piControl*. In contrast, the addition of freshwater in the North Atlantic in the *127kaFW* run results in a multi-millennial sustained cooling in the subsurface ocean near the Antarctic ice sheet. We explore the physical processes that lead to this new result and discuss implications for climate forcing of Antarctic ice sheet mass loss during the LIG.

## 1 Introduction

The Last Interglacial (LIG, 129 to 116 kyr ago [ka]) was characterized by warmer global temperatures and higher sea level compared to the pre-industrial (PI) climate, primarily because of orbitally-induced solar insolation anomalies. Reconstructions of the LIG from climate-proxy data indicate global mean temperatures of about 0.5–1.5°C greater than PI (Dutton et al., 2015; Kaspar et al., 2005; Otto-Bliesner et al., 2013; Fischer et al., 2018; Arias et al., 2021). Temperature anomalies were greatest near the poles, with Antarctic ice cores indicating annual air temperatures warmer by more than 2°C, peaking early during the LIG at 129–128 ka (Masson-Delmotte et al., 2011), and Greenland ice cores suggesting warming of more than 4°C (Dahl-Jensen et al., 2013; Landais et al., 2016; Masson-Delmotte et al., 2015). Sea-level records from the LIG indicate that global

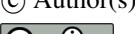



mean sea level was 4–9 m higher than present (Dutton et al., 2015). Geologic records indicate a smaller but intact Greenland ice sheet (Colville et al., 2011; de Vernal and Hillaire-Marcel, 2008). While some studies suggest that Greenland may have contributed as much as 5 m (e.g., Yau et al. (2016)), most recent studies have indicated that less than 2 m is more probable,

requiring large contributions from the Antarctic ice sheet (AIS), even for low-end estimates of global sea level rise (Dutton et al., 2015). The West Antarctic Ice Sheet (WAIS), which is grounded below sea level, is the most likely source of such contributions, and various lines of evidence suggest at least partial collapse of the WAIS occurring during the LIG (Scherer et al., 1998; Steig et al., 2004; Lau et al., 2023)

The canonical explanation for a large AIS contribution to sea level during the LIG is that a reduction in the Atlantic Merid-

ional Overturning Circulation (AMOC) played a critical role in driving ice loss from the AIS, and possibly a collapse of the WAIS (Goelzer et al., 2016; Turney et al., 2020; Clark et al., 2020). As large Northern Hemisphere ice sheets melted at the beginning of the LIG, large amounts of freshwater entered the North Atlantic, suppressing deep-water formation and leading to a reduction of the AMOC (Böhm et al., 2015), allowing heat to accumulate in the Southern Ocean (i.e., the bipolar seesaw pattern (Stocker and Johnsen, 2003; Marino et al., 2015)). Warmer Southern Ocean temperatures, in turn, would have enhanced

the melting of the Antarctic ice sheet margin. A large freshwater discharge, associated with the the Heinrich-11 (H11) event, is known to have occurred a few thousand years prior to the LIG (Böhm et al., 2015), consistent with this idea. However, both the magnitude and duration of the freshwater discharge, and the consequent changes in the Southern Ocean, are uncertain. It is thus an open question whether the H11 event was a *necessary* condition for AIS mass loss during the LIG. An alternative hypothesis is that warming in the Southern Ocean, owing simply to the insolation anomalies during the LIG, may have been

sufficient to cause substantial WAIS collapse. This question has relevance to the future of the WAIS over the next few centuries, given that a freshwater discharge event comparable to that of H11 is highly unlikely to occur in the present-day climate.

The most important unknown for the AIS during the LIG – and any period when the WAIS may have collapsed – is the ocean thermal forcing (TF), the difference between the *in situ* water temperature and the *in situ* melting point of ice. Some ice-sheet modelling studies suggest that a relatively high TF threshold must be reached before WAIS collapse is likely. For

example, Sutter et al. (2016) found that ocean warming of at least 2–3°C compared to PI was a prerequisite for collapse, while DeConto and Pollard (2016) found that, even with the inclusion of sensitive ice-calving physics and atmospheric feedbacks, the WAIS does not collapse under simulated LIG conditions without ocean warming of 3°C. However, other studies suggest that widespread WAIS mass loss could be triggered even with smaller TF (1–2°C) given the right ice sheet conditions and enough time (Berdahl et al., 2023; Lipscomb et al., 2021; Garbe et al., 2020).

The purpose of this study is to better understand the ocean and atmospheric conditions manifested near the AIS with and without a freshwater forcing event under LIG orbital conditions. Specifically, we evaluate the degree of ocean warming that may have occurred near Antarctica, with and without Northern Hemisphere freshwater forcing. To do this, we examine two global climate model simulations run with the Community Earth System Model version 2 (CESM2). The first is a simulation of the LIG, where the primary forcing difference from the PI is the orbital configuration. This leads to a large positive NH

solar summer insolation anomaly. The second simulation is the same as the LIG run, except for the addition of a large (0.2 Sv), continuous freshwater forcing in the North Atlantic. Previous work evaluating the large-scale features of similar LIG runs

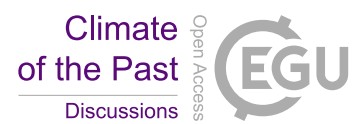

(e.g., Otto-Bliesner et al., 2020, 2021) showed that the large positive NH solar insolation anomaly results in summer warming over the NH continents and reduced Arctic summer minimum sea ice. Capron et al. (2017) evaluated the high-latitude surface climate during the *127ka* in global models as compared to proxy data. Here, we assess the state of the CESM2 pre-industrial and last interglacial climate with and without freshwater forcing. Our analyses focus largely on the Southern Ocean and Antarctic regions.

## 2 Model description

All simulations were run with the Community Earth System Model version 2 – Finite Volume 2 (CESM2-FV2x1), with approximately 2° resolution in the atmosphere and land models. Here we briefly summarize the components used in the CESM2 simulations; further details can be found in Danabasoglu et al. (2020). We use the relatively low resolution of FV2 in anticipation of running coupled experiments in future work, to save on computational expense. Otto-Bliesner et al. (2020) analyzed the *127ka* CESM2-FV1 simulation, which produces a global response similar to our FV2x1 simulation and complements this work.

CESM2 is a coupled Earth system model consisting of components for the ocean, atmosphere, sea ice, land, and land ice. Our simulations use the Community Atmosphere Model (CAM6) with 2.5° longitude by 1.8° latitude horizontal resolution. There are 32 levels, with the top level at 2.25 mb. The Community Land Model (CLM5) is on the same grid as CAM6. The ocean model is the Parallel Ocean Program version 2 (POP2). This is the same ocean model as was in the Community Climate System Model version 4 (CCSM4), except with some improvements in physical parameterizations. The POP2 resolution is a nominal 1° with uniform resolution of 1.125° in the zonal direction. The North Pole of the grid is displaced into Greenland, and there are 60 vertical levels with a maximum depth of 5500 m. The upper 16 levels have a uniform thickness of 10 m, while below 160 m, the level thicknesses increase monotonically to ∼3500 m depth. The deepest 2000 m have a nearly uniform thickness of ∼250 m extending to the ocean floor. The sea ice model (CICE5) uses the same grid as the ocean model. CESM2's land ice component, the Community Ice Sheet Model (CISM2.1, Lipscomb et al. (2019)) typically runs on a regular 4 km mesh and is available in a coupled framework with CESM2 for the Greenland ice sheet (GrIS). Coupling to Antarctica is still in the testing phase. In the simulations presented here, the GrIS is not evolving and its geometry remains fixed. However, both surface temperature and surface mass balance (SMB) are computed in CLM5 and downscaled to the finer CISM grid. Similarly, the AIS is not evolving; the SMB is computed in CLM5 but not downscaled to a finer CISM grid.

### 2.1 Experimental design and methods

We compare two simulations using CESM2 under different 127 ka scenarios. The first is a 1000-year control simulation at 127 ka. This simulation, herein referred to as *127ka*, used the CMIP6–PMIP4 Tier 1 protocol. The simulation assumes modern continental configuration and prescribes orbital parameters and greenhouse gases to LIG levels (Otto-Bliesner et al., 2017). Potential vegetation, which removes crops and urban areas and replaces it with suitable natural vegetation types, is used rather than the 1850 CE pre-industrial (PI) vegetation (Otto-Bliesner et al., 2020). The most notable change in forcing during the

*127ka* compared to PI is orbital. The orbit at *127ka* is characterized by larger eccentricity and tilt than PI, and perihelion

occurs close to the boreal summer solstice as compared to near the boreal winter solstice during the PI. This results in a large positive Northern Hemisphere (NH) solar insolation anomaly from April to September. Similarly, there are negative insolation anomalies in December–January–February (DJF). Full details on the *127ka* orbital forcing can be found in Otto-Bliesner et al. (2020). Other forcings and boundary conditions such as the solar constant, ozone, volcanic aerosols, topography, and ice sheet configuration remain the same as in the PI.

The second run, *127ka-FW*, is a CMIP6-PMIP4 Tier 2 simulation that uses the same configuration and forcing as *127ka*, except for a large meltwater flux (0.2 Sv) added at the surface of the North Atlantic from 50-70°N. This forcing is meant to simulate iceberg rafting from ∼135 to 130 ka as Northern Hemisphere ice sheets retreated at the end of the penultimate glaciation (Marino et al., 2015). Although this experiment was originally designed to simulate the H11 event (Otto-Bliesner et al., 2020), H11 actually began several thousand years before the LIG, during the penultimate deglacation and the response

to freshwater forcing in climate models is quite different in colder glacial versus warmer interglacial states (Bitz et al., 2007). We therefore use the nomenclature *127kaFW*, rather than *127-H11*. This simulation was originally run for 1000 years under the PMIP4 protocol. We extend it with continuous freshwater forcing for another 3000 years, for a total of 4000 years. This multi-millennial simulation is designed to allow the ocean to adjust to the freshwater forcing. Although not fully realistic in terms of timing, this simulation provides a useful point of comparison with the *127ka* run (without freshwater forcing), as well

as with the results of earlier work in which freshwater forcing begins at 138 ka (e.g., Clark et al., 2020).

We also compare the two LIG simulations to a CMIP6 Diagnostic, Evaluation, and Characterization of Klima (DECK) pre-industrial (PI) control run, *piControl*. This is a 1000-year control simulation under pre-industrial climate forcings. Table 1 shows more details for all three simulations.

## 3 Results

In Section 3.1 we analyze the mean climate of the *piControl*. We then compare the climate of the LIG (*127ka*) to the PI (*piControl*), primarily focusing on Antarctica and the Southern Ocean. All anomalies are computed as the difference between 127ka and PI (difference = *lig127- piControl*). Climatologies are computed using the final 50 years of the simulations. Section 3.2 examines the impact of the freshwater forcing on the climate response in the Southern Ocean, and the implications for AIS mass loss.

### 3.1 Evaluating the simulated PI and LIG climates (*piControl & 127ka*)

#### 3.1.1 Global assessment

As in Otto-Bliesner et al. (2020), we find that global mean annual anomalies in surface air temperature in the *127ka* simulation are slightly positive compared to the PI simulation. Global mean surface air temperature in CESM2 (*127ka*) is only 0.004°C warmer than the PI (*piControl*) (Table2). While global JJA temperatures are warmer in the LIG simulation by about 1°C, global



| Parameter | *piControl* | *127ka* | *127kaFW* |
|---|---|---|---|
| orbit | 1850 | 127ka | 127ka |
| solar | Fixed SSI, 1850-1873 mean | Same as *piControl* | Same as *piControl* |
| $CO_2$ (ppm) | 284.7 | 275.0 | 275.0 |
| $CH_4$ (ppb) | 791.6 | 685.0 | 685.0 |
| $N_2O$ (ppb) | 275.68 | 255.0 | 255.0 |
| Other GHG (CFCs) | DECK *piControl* | 0. | 0. |
| Ozone | DECK *piControl* | Same as *piControl* | Same as *piControl* |
| Volcanic aerosols | Background 1850-2014 mean from WACCM ensemble | Same as *piControl* | Same as *piControl* |
| Aerosols (excluding dust) | DECK *piControl* | Same as *piControl* | Same as *piControl* |
| Vegetation | 1850 CE PI Vegetation and crops | Potential Vegetation, no urban, no crops | Potential Vegetation no urban, no crops |
| Land Surface Topography | Modern | Modern | Modern |
| Ice Sheets | Modern | Modern | Modern |
| Heinrich Freshwater Forcing | None | None | Continuous 0.2 Sv between 50-70°N |
| # spinup years | 1,070 | 325 | 325 |
| # production years | 1000 | 1000 | 4000 |

**Table 1.** Summary of CESM2-FV2 forcings and boundary conditions used in the *127ka*, *127kaFW* and *piControl* runs.

| | Annual | | | | DJF | | | | JJA | | | |
|---|---|---|---|---|---|---|---|---|---|---|---|---|
| | Global | NH | SH | Antarctic | Global | NH | SH | Antarctic | Global | NH | SH | Antarctic |
| 127ka | 14.093 | 15.02 | 13.17 | -29.36 | 11.49 | 7.72 | 15.27 | -18.15 | 16.79 | 22.54 | 11.03 | -38.15 |
| PI | 14.089 | 15.05 | 13.13 | -29.75 | 12.50 | 9.03 | 15.97 | -16.65 | 15.59 | 20.64 | 10.55 | -38.54 |
| 127-PI | 0.004 | -0.03 | 0.04 | 0.39 | -1.01 | -1.31 | -0.7 | -1.502 | 1.0 | 1.9 | 0.48 | 0.38 |

**Table 2.** Climatological 2m air temperature (°C). Climatology defined as last 50 years of the run.

DJF temps are cooler by a similar magnitude. This is consistent with the timing of insolation anomalies in NH summer and winter (Otto-Bliesner et al., 2020). The greatest near-surface air temperature anomaly occurs in the NH during JJA, reaching nearly 2°C. Proxy evidence suggests global temperatures were roughly 0.5–1°C warmer during the LIG than PI (Arias et al., 2021; Dutton et al., 2015), indicating that the orbital-only forcing underestimates global average temperature anomalies. However, errors associated with the proxies themselves can often be as large as the proxy estimate of the differences between

the LIG and PI (Fig. A2(e)). Otto-Bliesner et al. (2020) likewise note that CESM2 simulates the positive anomaly patterns in Greenland and Antarctica, but underestimates the magnitude of the reconstructed anomalies.





The global ocean heat content increases during the *127ka* simulation. Sea surface temperatures show significant regional variability, with the strongest anomalies occurring in the North Atlantic (cooling) and the Southern Ocean (warming) (Fig. A3). Subsurface ocean temperatures in the Atlantic basin cool during the *127ka* simulation, whereas most other basins warm,

including the Pacific, Arctic and Southern Ocean. This is consistent with analyses by Otto-Bliesner et al. (2020), which showed that stronger westerlies over the North Atlantic increase the magnitude of the wind stress curl in the vicinity of the North Atlantic subpolar gyre and the western boundary of the subtropical gyre. Global and Atlantic northward heat transport is generally greater at all latitudes in the *127ka* simulation compared to the *piControl* (i.e., more vigorous transport, not shown). The AMOC strengthens, as indicated in both the upper NH cell and deeper SH cell during *127ka* compared to *piControl* (∼2 Sv

increase). This is consistent with cooler North Atlantic sea surface temperatures (Fig. A3), as deep water formation increases and dense surface waters are cooled and brought to depth. Otto-Bliesner et al. (2020) provides more detail on the barotropic streamfunction and AMOC changes during *127ka*.

Seasonal sea ice climatologies indicate that summer NH sea ice area is reduced by ∼5 million km$^2$ during *127ka* as compared to the PI (Fig. A4). Arctic sea ice loss is widespread across the basin during *127ka* as a result of increased NH summer

insolation. Notably, the Arctic is almost ice-free in summers, with only some thin sea ice remaining in September in the central Arctic.

For more analyses on global climate changes during *127ka* (e.g., precipitation/monsoon patterns, El Niño Southern Oscillation (ENSO), and AMOC), we refer the reader to the FV1 analyses by Otto-Bliesner et al. (2020).

### 3.1.2   Antarctic assessment

Many CMIP models suffer from a warm sea surface and deep ocean bias in the Southern Ocean (Luo et al., 2023). CESM2 is one of these models with a global-mean SST warm bias (Danabasoglu et al., 2020). The PI Southern Ocean below ∼200 m depth is warmer than the World Ocean Atlas observations by 1–2°C (not shown). Comparisons of ocean temperature and salinity depth profiles in the Amundsen Sea indicate that the model captures the salinity structure of the region fairly well, but with a warm bias of ∼1°C below the thermocline (Fig. A1).

The general climate state of the *piControl* run, including air temperatures, ocean temperatures at the surface and subsurface, near-surface winds, wind stress curl, and minimum (Feb) sea ice extent, are shown in Fig. 1. Notably, the mean ocean temperature (MOT) of the full column in the SO is between 1–2°C, while MOT in the 200–800m range is warmer than this by about 0.5°C. Mean westerly winds induce a negative wind stress curl in the SO south of 50°S. Minimum sea ice extent in *piControl* is about 3.03 million km$^2$, comparable to the the median observed extent (1981-2010) of 2.99 million km$^2$ (NSIDC).

Mean Antarctic near-surface air temperatures are shown in Table 2. Proxy reconstructions indicate that the Southern Ocean and Antarctic surface air temperatures at 127ka were 1.8 ± 0.8°C and 2.2 ± 1.8°C warmer, respectively, than PI conditions (Capron et al., 2017). Our orbital-only *127ka* simulation generates Antarctic annual surface air temperatures roughly 0.4°C warmer than *piControl* (Fig. 2). In DJF (austral summer), simulated air temperatures over Antarctica are about 1.5°C cooler in *127ka*, while in JJA (austral winter), temperatures are 0.38°C warmer (Table 2). The largest difference occurs in austral spring

(Sep-Nov), when *127ka* is about 2.7°C warmer than *piControl*.







**Figure 1.** Climatological *piControl* 2m air temperature (a), near surface ocean temperature (b), mean ocean temperature (MOT) for the full ocean column (c), MOT for 200-800m depth range only (d), surface winds and wind stress curl (e), and February (minimum) sea ice extent (f). Climatologies are computed over the last 50 years of the simulation.





While in general, temperatures over the Antarctic in our simulations are lower than the proxy data from Capron et al. (2017) suggest, we note that recent estimates of the sensitivity of water isotopes to temperature for East Antarctica suggests considerably smaller glacial–interglacial surface-temperature changes in East Antarctica than previously thought (e.g. Kahle et al., 2021; Buizert et al., 2021). This same scaling, if applied to the LIG, also suggests much smaller warm anomalies than earlier work suggests. In particular, recent work suggests that the peak interglacial warmth at Dome Fuji, central East Antarctica was only 2°C warmer than present (Oyabu et al., 2023), as opposed to earlier estimates of 3.3°C (Capron et al., 2017). Furthermore, that work suggests that the peak interglacial warmth occurred at about 129 ka, and warming at 127 ka was probably only about half as large, perhaps less than 1°C. Given this emerging new work, Antarctic temperatures in our *127ka* simulation may not be inconsistent with proxy data (Figure 2a), though are probably on the low end.

Southern Ocean SSTs are warmer on average in *127ka* compared to *piControl*, with some regional cooling such as in the Weddell, Amundsen, and Ross Seas (Fig. 2). The subsurface Southern Ocean is generally warmer during *127ka*, including all subset regions of the high-latitude Southern Ocean (ocean regions south of the pink band, Fig. 3). Table 3 breaks down the regional ocean response depth in the top 1500 m. In the Southern Ocean (ocean regions south of the black band in Fig. 3), ocean temperatures increase at all levels, with anomalies from PI ranging from 0.1°C to 0.25°C in the upper 1500 m of the ocean. Closer to the continent in the tight Southern Ocean (TightSO, south of 65°S), subsurface temperature anomalies in the upper 1500 m range from 0.15°C to 0.27°C compared to *piControl*. TightSO SST anomalies during *127ka* are generally negative, apart from in the East Antarctic Ice Sheet (EAIS) region, though these changes are tied to changes in sea ice. The large warm air temperature patch (Fig. 2a,b) largely accounts for the positive SST anomaly in the EAIS sector (Fig. 2d). Temperature anomalies in the TightSO reach around 0.25°C below 500 m. Notably, subsurface anomalies reach at least 0.23°C in all TightSO sectors. The largest anomalies occur in the Tight Weddell below 200 m, with anomalies reaching close to 0.4°C.

The *127ka* run simulates a reduction in Antarctic maximum sea ice extent, and a slight increase in minimum Antarctic ice area (Fig. A4). Expansion of minimum Antarctic sea ice is evident around much of the continent, except for a reduction in extent in the EAIS near Dronning Maud Land (Fig. 4a). *127ka* simulates slightly higher snowfall rates on average across the continent as compared to the *piControl* ($\sim$ 1.72 mm/yr), (Fig. 4b). Regionally, climatological snowfall across the continent indicates a reduction of coastal snowfall in the Amundsen, Dronning Maud Land and Wilkes Land under LIG solar insolation as compared to the PI. Meanwhile, the Antarctic Peninsula (AP), Enderby Land, and the Ross and its neighboring regions have more snowfall. The interior of the ice sheet has a mixture of higher and lower snowfall during the LIG, but magnitudes are small, and generally differences in the interior are statistically insignificant.

As noted, the Southern Ocean generally warms as a result of the orbitally-forced *127ka*. MOT anomalies from the full ocean column show subsurface warming, with the largest temperature anomalies of about 0.4°C in the Weddell Sea region (Figure 4c). At depths near the average grounding lines of most Antarctic ice shelves ($\sim$ 200–800 m) – depths that are most relevant to sub-shelf melt rates – MOT anomalies are even higher (Fig. 4d). This warming remains robust across seasonal means as well (Table A1).

In the Southern Ocean, Otto-Bliesner et al. (2020) showed that in the *127ka* FV1 simulation the Antarctic Circumpolar Current (ACC) weakens during the LIG, consistent with a small shift northward of the southern westerlies in the Pacific sector.





**Figure 2.** Top row: Climatological *127ka* Antarctic 2m air temperature anomaly for annual, JJA and DJF seasons. Bottom row: Climatological *127ka* ocean temperature anomaly for near surface, ∼500m and ∼1500m ocean depths. Anomalies are with respect to *piControl*. Climatologies are computed over the last 50 years of each simulation. Proxy data indicating anomalies between 127ka and PI are overlain where available in the region (Data from Otto-Bliesner et al. (2020); Capron et al. (2017). The ice-core-based estimates of Antarctic surface temperature are scaled following the isotope-temperature scaling in Kahle et al. (2021) and Buizert et al. (2021)

.





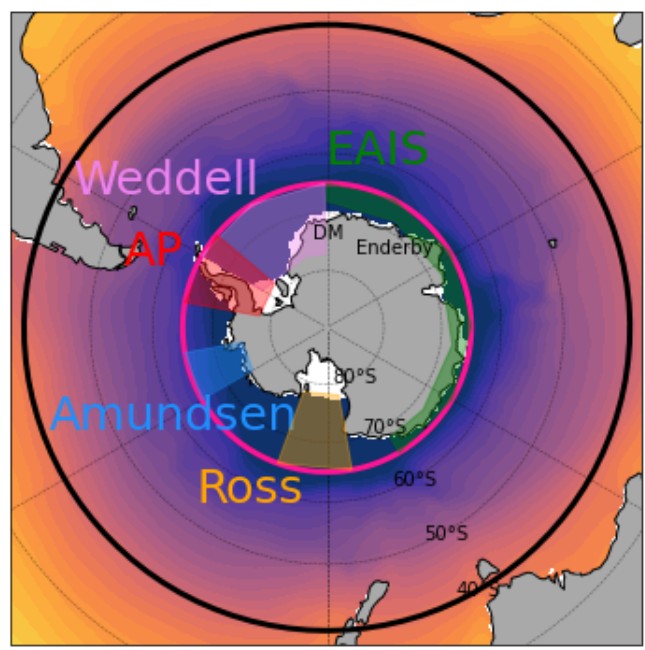

**Figure 3.** Regions used in masking around Antarctica. EAIS and AP are abbreviations for East Antarctic Ice Sheet and Antarctic Peninsula, respectively. DM indicates Dronning Maud Land, Enderby refers to Enderby Land. Ocean regions south of the black band denote the Southern Ocean, and south of the pink band (65°S) denote the Tight Southern Ocean. Colored regions show sectors used in Table 3.

The greatest weakening of westerlies during the LIG is in the Atlantic–Indian Ocean sector (Fig. 4e). This region coincides with decreasing minimum sea ice extent and the greatest ocean warming both at the surface and throughout the ocean column, as seen in the MOT (Fig. 4c). The anomalies in wind stress curl around Antarctica (Fig. 4f) show lower magnitude wind stress curl (less negative, see Fig. 1 for mean state reference) near the continent (red shades) during *127ka* than in *piControl*. Areas of

negative wind stress curl in the SH indicate surface water is deflected northward and replaced by upwelled water from below (ie. positive Ekman suction). In *127ka*, close to the continent (the TightSO), curl anomalies are generally positive, indicating Ekman suction is stronger in the PI than the LIG. In other words, there is less upwelling in the LIG than in PI. North of the TightSO boundary, the opposite is true. This mean annual pattern is dominated by DJF rather than JJA.

### 3.2 Model Response to Freshwater Forcing

Next, we analyze the impact of the synthetic freshwater forcing in the North Atlantic. In the following analyses, all anomalies are calculated with respect to the *127ka* run (i.e., *127kaFW - 127ka*), unless indicated otherwise.





**Figure 4.** Climate anomalies between *127ka* and *piControl*. Climatological *127ka* anomaly of minimum (March) sea ice concentration (a), snowfall rate (hatching indicates significance) (b), mean ocean temperature (MOT) for the full ocean column (c), MOT (200-800m only) (d), surface winds (e) and wind stress curl (f). Anomalies are with respect to *piControl*. Climatologies are computed over the last 50 years of each simulation.





|  | Southern Ocean | | | Tight Southern Ocean | | | Tight Amundsen | | |
|---|---|---|---|---|---|---|---|---|---|
|  | 127ka | piControl | Diff | 127ka | piControl | Diff | 127ka | piControl | Diff |
| SST | 7.98 | 7.88 | 0.1 | -1.54 | -1.49 | -0.05 | -1.35 | -1.16 | -0.19 |
| 200m | 6.36 | 6.11 | 0.25 | 1.26 | 1.12 | 0.15 | 1.85 | 1.61 | 0.23 |
| 500m | 4.81 | 4.59 | 0.22 | 1.95 | 1.7 | 0.24 | 2.41 | 2.22 | 0.19 |
| 750m | 3.77 | 3.60 | 0.17 | 1.9 | 1.67 | 0.24 | 2.23 | 2.05 | 0.18 |
| 1000m | 3.03 | 2.88 | 0.16 | 1.79 | 1.52 | 0.26 | 1.96 | 1.77 | 0.19 |
| 1500m | 2.73 | 2.57 | 0.16 | 1.66 | 1.4 | 0.27 | 1.81 | 1.60 | 0.21 |

|  | Tight AP | | | Tight Ross | | | Tight Weddell | | | Tight EAIS | | |
|---|---|---|---|---|---|---|---|---|---|---|---|---|
|  | 127ka | piControl | Diff | 127ka | piControl | Diff | 127ka | piControl | Diff | 127ka | piControl | Diff |
| SST | -1.53 | -1.47 | -0.06 | -1.5 | -1.46 | -0.07 | -1.74 | -1.7 | -0.03 | -1.46 | -1.59 | 0.13 |
| 200m | 0.53 | 0.37 | 0.16 | 1.28 | 1.12 | 0.16 | 0.72 | 0.59 | 0.14 | 0.95 | 0.93 | 0.02 |
| 500m | 0.99 | 0.76 | 0.23 | 1.63 | 1.44 | 0.19 | 1.82 | 1.43 | 0.38 | 1.98 | 1.744 | 0.24 |
| 750m | 1.06 | 0.86 | 0.2 | 1.71 | 1.49 | 0.22 | 1.86 | 1.52 | 0.34 | 1.87 | 1.63 | 0.24 |
| 1000m | 0.98 | 0.82 | 0.16 | 1.63 | 1.37 | 0.26 | 1.77 | 1.46 | 0.31 | 1.69 | 1.43 | 0.26 |
| 1500m | 0.88 | 0.74 | 0.14 | 1.54 | 1.25 | 0.29 | 1.66 | 1.37 | 0.3 | 1.53 | 1.27 | 0.26 |

**Table 3.** Regional climatological ocean temperatures (°C). Columns denoting differences are defined as (*127ka - piControl*). Climatology is defined as the last 50 years of each run. Regions correspond to those shown in Fig. 3.

### 3.2.1 Global Assessment

As expected, the addition of freshwater to the North Atlantic in *127kaFW* causes the AMOC to quickly collapse and global temperatures to drop. The AMOC remains suppressed by $\sim 75 - 80\%$ compared to *127ka* over the course of the experiment ($\sim$4000 years) (Fig. 5a). The AMOC suppression is also seen as a reduction in northward heat transport in the Atlantic by $\sim 65\%$ (Fig. 5b). Global air temperatures decrease by more than 2°C in the first few centuries, following by slower global cooling for the following several millennia (Fig. 5c). The reduction in Atlantic northward heat transport induces the robust bipolar seesaw response (NH cooling and SH warming) in air temperatures (Fig. A2c) and sea surface temperatures (Fig. 5d). Sea surface temperatures in the SH warm by over half a degree on average during the first 500 years as a result of the freshwater forcing, after which warming continues but at a reduced rate. NH SSTs, by contrast, cool more than 2°C in the first century. SSTs continue to cool at a slower rate after the initial shock. By the end of 4000 years, NH SSTs cool by over 3°C compared to the *127ka* simulation.

Arctic sea ice expands significantly as a result of the freshwater forcing, almost doubling from the initial $\sim$11 million km$^2$ to $\sim$22 million km$^2$ by the end of the simulation (Fig. 5e). The majority of change (i.e., an increase in sea ice extent of $\sim$9 million km$^2$) occurs within the first few centuries, followed by a slow and steady increase for the next 3500 years. The increase in extent during *127kaFW* occurs in all months, rather than being dominated by a certain season (Fig. A4). Seasonal minimum



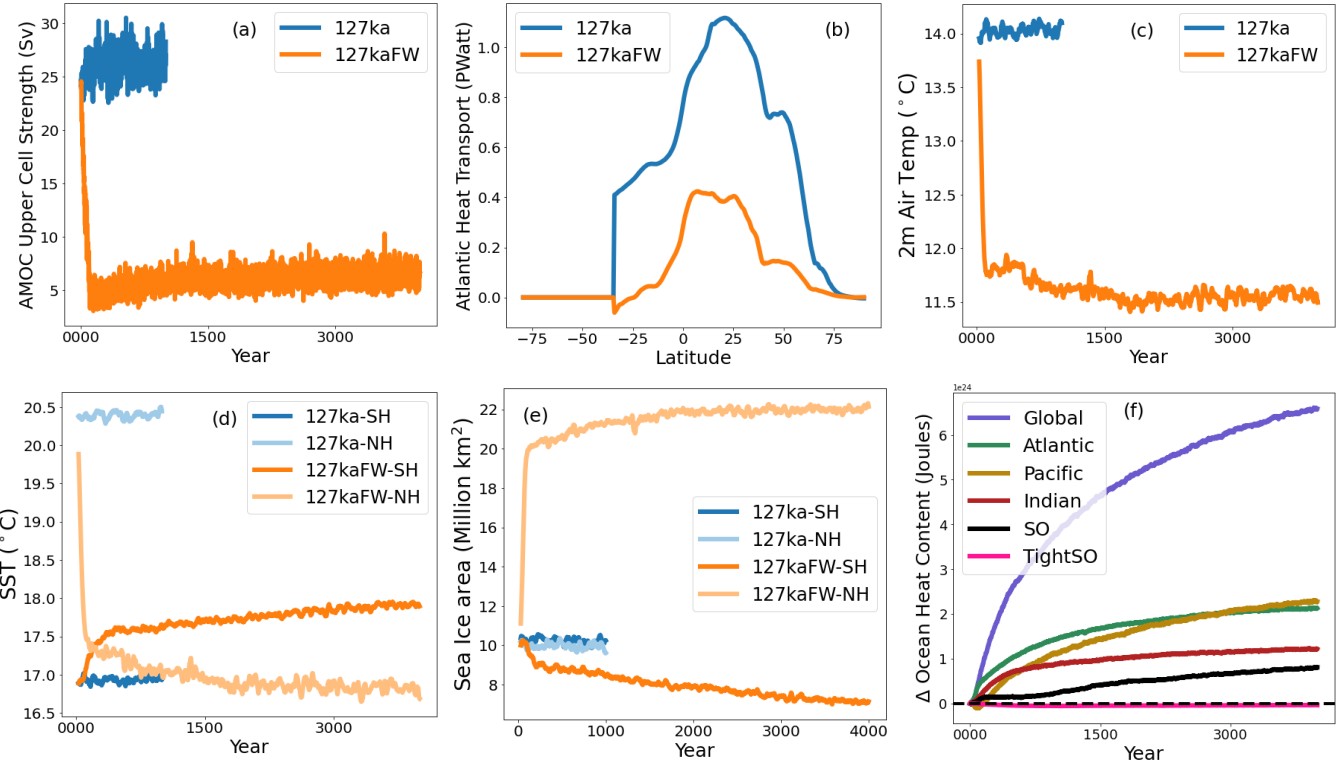

**Figure 5.** Global response to *127kaFW* forcing. (a) North Atlantic annual upper cell strength (calculated as the maximum streamfunction found north of 28°N, and below 500 m depth. (b) Atlantic northward heat transport. 127kaFW shows the average for the final 100 years of the simulation (year 3900-4000), though this pattern is representative of the full simulation after the initial shock. The *127ka* is the climatological mean (final 50 years of the simulation). (c) Global mean 2m surface air temperatures, 30-year rolling mean. (d) NH and SH SSTs for the *127kaFW* and *127ka* runs. Temperatures computed as 30-year rolling mean.(e) SH and NH sea ice area (million km$^2$) for *127ka* and *127kaFW*, 30 year running means. (f) Change in ocean heat content during the *127kaFW* simulation for global ocean and specific basins. Dashed black line indicates zero change.

Arctic sea ice extent during *127kaFW* is comparable to the maximum seasonal extent in the *127ka* run (∼15 Million km$^2$). Antarctic sea ice, on the other hand, diminishes quickly in the first several centuries of the freshwater experiment, and thereafter continues to steadily decline, reaching ∼7 million km$^2$ by the end of the simulation, a reduction of ∼ 30% area. The magnitude
of Antarctic sea ice extent change is much smaller than that in the Arctic during *127kaFW*. The *127ka* simulation indicates reduction of Antarctic sea ice in all seasons compared to both the *127ka* and *piControl*.

In the freshwater experiment, the reduction of overturning in the Atlantic leads to heat accumulation in all of the main global ocean sectors (Atlantic, Pacific, Indian, Southern Ocean, Fig. 5f). However, close to the Antarctic continent in the TightSO (Fig. 3), there is a sustained reduction in ocean heat content for the duration of the *127kaFW* simulation. We elaborate on the
TightSO cooling in the next section.



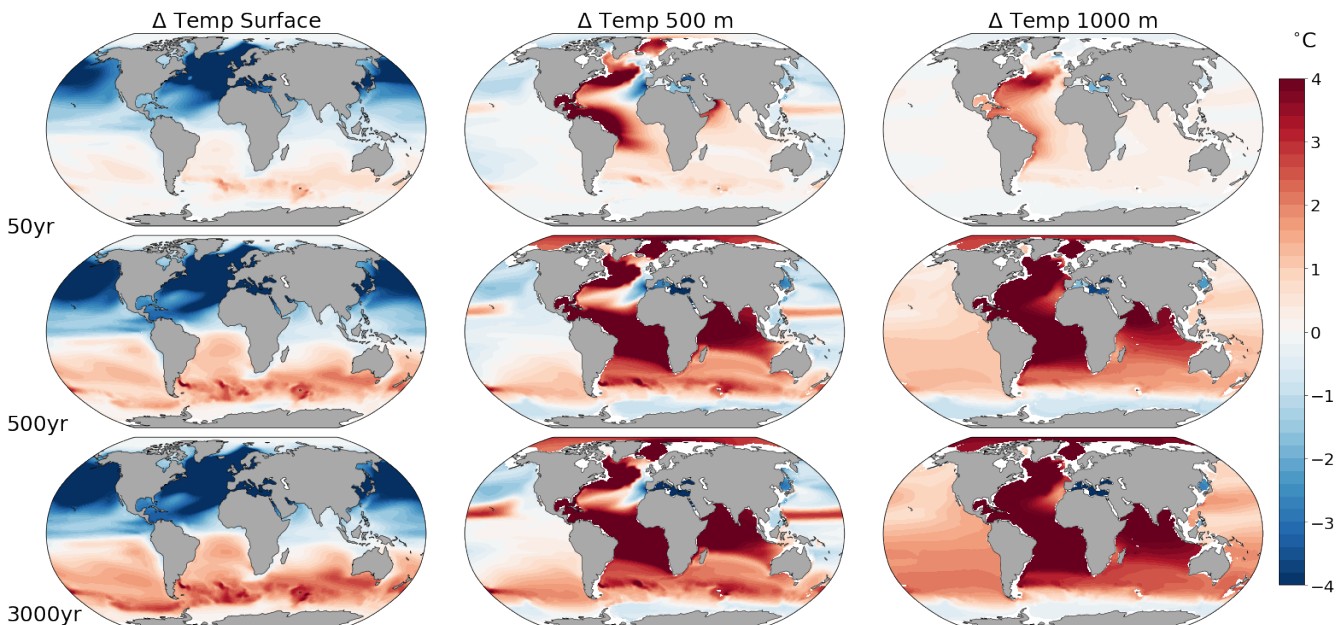

**Figure 6.** Ocean temperature anomalies for *127kaFW* relative to *127ka* for three time slices (rows) and three depths (columns). Differences show the climatological difference between *127kaFW* and *127ka*, where climatology is computed as the final 50 years of the each simulation.

The global SST spatial pattern resulting from the freshwater forcing is shown in Fig. 6. Northern Hemisphere SST cooling and Southern Hemisphere warming is a robust response to the addition of freshwater in the North Atlantic. North Atlantic Deep Water formation is reduced due to modification of density-driven circulation in the North Atlantic. Subsurface waters, however, indicate a more complex response. The freshwater forcing causes notable ocean warming at 500 m ($> 4°C$), particularly in the
Atlantic, Indian, and Arctic Oceans. At this depth, cooling occurs in the North and Equatorial Pacific, as well as the Southern Ocean near the Antarctic continent. At 1000 m depth, almost the entire global ocean warms in response to the freshwater forcing. The Southern Ocean, however, shows distinct cooling at depth. In the next section, we explore the mechanisms and implications for this reduction of ocean heat near the AIS.

### 3.2.2 Antarctic Assessment

As noted earlier, global surface air temperatures fall significantly during *127kaFW* as compared to the *127ka* run. Freshwater forcing causes global annual average temperatures to decrease by more than $2°C$, with DJF temperatures falling by $\sim3°C$ and JJA temperatures by $\sim1.5°C$ (Fig. A5). Over the Antarctic continent, however, surface air temperatures increase as a result of the freshwater forcing. There appear to be two timescales of air temperature response in the SH. In the initial rapid response, annual Antarctic air temperatures increase by $\sim1°C$ within the first several centuries. This is followed by a more moderate
but steady increase in temperatures for the rest of the simulation, reaching about $2.5°C$ warming after 4000 years (Fig. 7a).



Antarctic summer (DJF) near-surface air temperatures increase by almost 2°C and winter (JJA) temperatures by ∼3°C (Fig. A5).

**Figure 7.** Antarctic and Southern Ocean response to *127kaFW* forcing. All curves show 30-year rolling means. (a) Near-surface air temperature over the Antarctic continent. (b) Global meridional overturning circulation (MOC) lower cell strength. Location extracted where minimum streamfunction is located, below 2km depth and south of 33°S. Larger negative values indicate stronger circulation (negative indicates counterclockwise flow). (c) Mean ocean temperature (MOT) for the full ocean column in different sectors near the continent (sectors follow Fig. 3). (d) Same as (c) but for depths from 200–800 m.

Results in the previous section indicated that the Southern Ocean accumulated heat in response to the idealized freshwater forcing in the model. However, closer to the AIS, in the TightSO, we found a sustained cooling. This has important implications for ocean thermal forcing on the ice sheet, and therefore ice sheet melt rates and mass loss. Near the Antarctic continent (TightSO regions, Fig. 3), we find that the MOT of the full ocean column decreases by up to 0.25°C over the first millennium

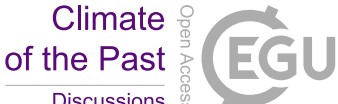

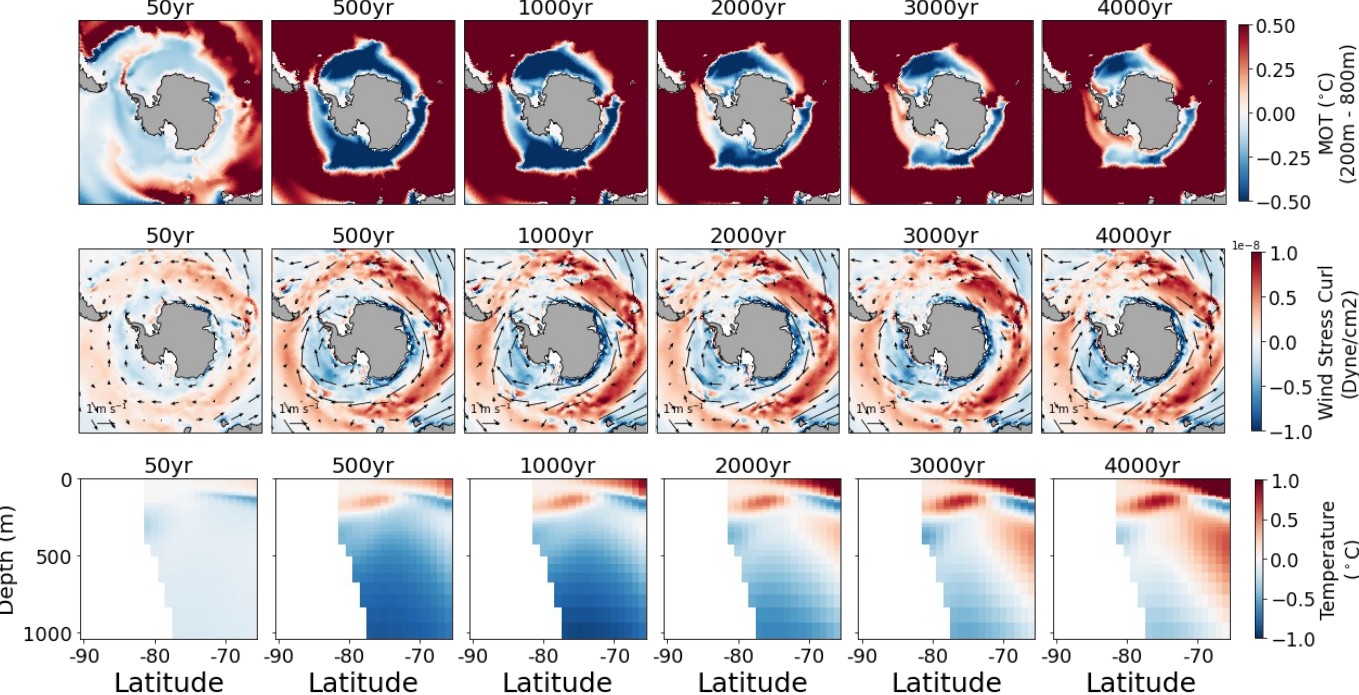

**Figure 8.** Anomaly between *127kaFW* and climatological *127ka* near Antarctica. (a) Mean ocean temperature from 200–800 m depth. (b) Wind stress curl (shading) overlain with surface winds. (c) Ocean temperature with depth cross-section off the Amundsen coast. Cross-section transect location shown in panel (a). 50-year average snapshots are shown for years 50, 500, 1000, 2000, 3000 and 4000 of the freshwater simulation.

(Fig. 7c). After this multi-century cooling, MOTs rebound slowly for the remainder of the simulation. Importantly, MOTs in the TightSO on average, as well as in some regional subsets of the TightSO, never rebound back to their initial values. Given that the AIS is most sensitive to thermal forcing near the grounding line of the ice shelves, we also compute the MOTs just

for these depths (200m-800m) (Fig. 7d), and find similar results. On average in the TightSO, at depths most relevant to the ice sheet, temperatures cool by ∼0.2°C within the first millennium (Figure 7d). In certain sectors, MOTs cool even more (up to 0.25°C in the EAIS and Ross), and again rebound slowly after the initial cooling. On average in the TightSO, as well as in the EAIS and Ross regions, it takes almost the full 4000 years to reach initial MOTs in this depth range. In the Amundsen and AP regions, MOTs rebound the most quickly after the initial cooling, exceeding their initial values after 1500 years. The Weddell

sector never rebounds to its initial values before the end of the experiment. While this cooling response in the subsurface occurs in other freshwater forcing experiments (e.g., He et al. (2021); Pedro et al. (2018)), it has not been considered or explored in the literature, nor has it been considered in the context of implications for AIS mass loss.

The spatial patterns of MOT further illustrate the timing of regional cooling and warming in response to the freshwater forcing (Fig. 8, top row). Within the first 500 years, a bulwark of cold subsurface water wrapping around the continent is





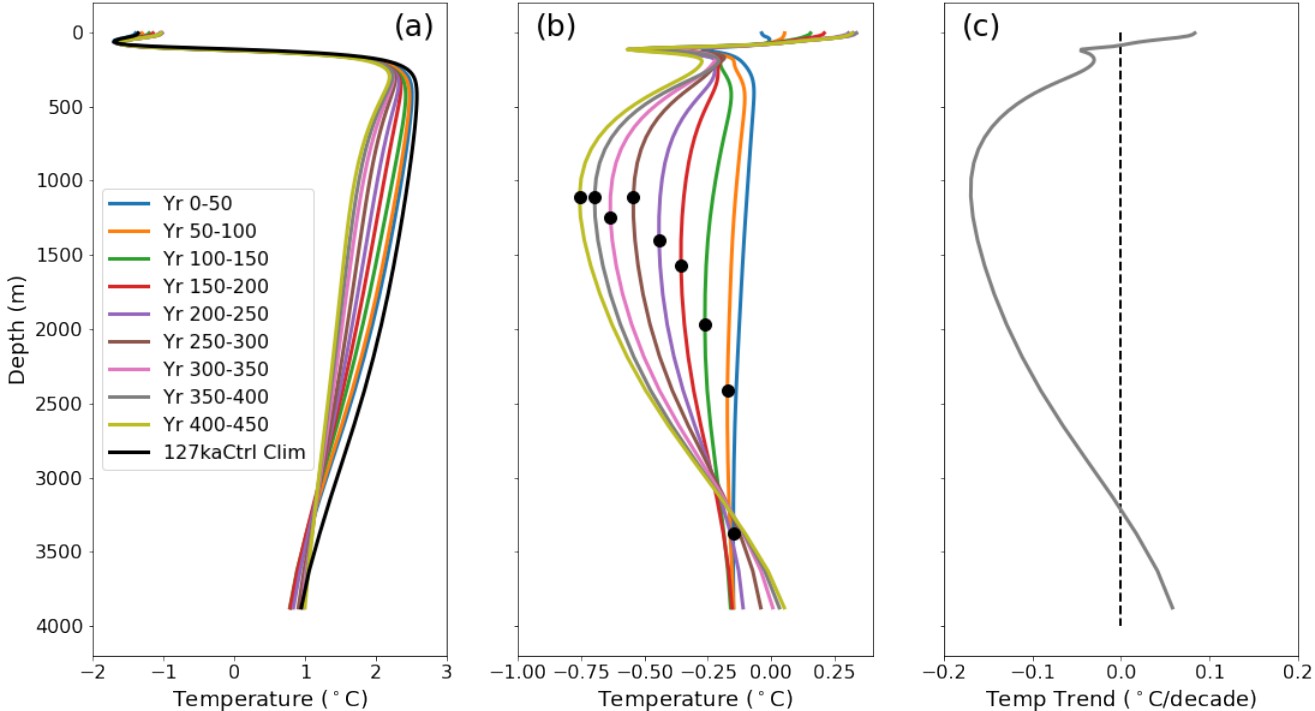

**Figure 9.** Evolution of ocean thermal structure off the Amundsen Sea continental shelf (110°W, 68.5°S). Climatological temperature profile (black curve) for *127ka* compared to 50 year means for the first 500 years (colors) (Panel a). Panel (b) shows anomalies with respect to the *127ka* control run. Black dots indicate the minimum temp below the thermocline. Panel (c) shows decadal temperature trend over the first 500 years. Dashed line indicates zero temperature trend.

established. Warm water propagates closer to the continent over the next few thousand years in a heterogeneous spatial pattern. The first region to encounter warm anomalies is the Amery, about 1000 years into the simulation. This is followed by warming along the peninsula after ~2000 years, and then warming in the Amundsen and Western Ross Sea after ~3000 years. The cold anomaly persists to the end of the simulation in the Weddell Sea.

Other work has explored the thermal bipolar seesaw response to freshwater forcing (e.g., Guarino et al. (2023); Clark et al. (2020)). Pedro et al. (2018) ran AMOC collapse experiments using CCSM3 (CESM's predecessor) to show that the heat reservoir during collapse events is actually north of the Atlantic Circumpolar Current (ACC), and not the oft-assumed Southern Ocean. They argue that eddy transport is necessary to move heat south across this dynamic barrier, which operates with a lag compared to the initial wind-driven response in ocean properties. To tease apart the mechanisms driving the initial subsurface cooling in the TightSO, as well as the subsequent rebound in subsurface ocean temperatures nearest the AIS, we examine the shift in winds and eddy transport across the ACC.

Generally, peak westerly winds are known to cause divergent surface flow that draws water up from below in a broad ring circling the continent (Morrison et al., 2015; Armour et al., 2016). This upwelling is well-captured in the *127ka* climatology



(Fig. A6). The sudden AMOC reduction and consequent changes in heat redistribution in *127kaFW* cause changes in atmospheric circulation. Anomalous surface winds impart wind stress at the ocean surface, affecting ocean upwelling rates. We find

the peak Southern Hemisphere westerlies shift south by several degrees in the first few hundred years following freshwater forcing, and remain in their poleward-shifted position for the rest of the simulation. As the westerlies move southward, a band of negative wind stress curl anomalies encircles the continent (blue shades, Fig. 8, second row). In the Southern Hemisphere, negative wind stress curl anomalies enhance the upward vertical velocities in the subsurface ocean (A6).

In order to visualize the initial subsurface cooling in the Amundsen Sea, we extract a cross section of the upper 1000 m

of the water column (Fig. 8, bottom row). These cross-sections show that initially there is warming confined to the upper 150–200 m of the column, below which the ocean cools. Again, we see that maximum subsurface cooling in response to the freshwater forcing is reached within the first millennium. Further examination of the evolution of the ocean thermal structure during the initial 500 years of *127kaFW* at a location off the shelf in the Amundsen Sea (110°W, 68.5°S) reveals warming above the thermocline and cooling below (Fig. 9a). Peak cold anomalies propagate upwards over time (black dots, panel (b)).

The cooling trend peaks at around 1 km depth (Fig. 9c).

After the first millennium, warm CDW moves toward the continent, reaching depths where ice shelf grounding lines are located (Fig. 8, top/bottom row). After about 3000 years, Amundsen Sea ocean temperatures near the AIS experience a net warming with respect to *127ka*. Warming continues for the final 1000 years of the simulation.

Our simulations also suggest that the eddy component of the global MOC in the deep Southern Ocean intensifies quickly

during the first millennium of the simulation and remains more vigorous (∼50% increase) for the remainder of the simulation (Fig. 7b). In other words, the eddy-driven transport in the SO strengthens in the *127kaFW* simulation. This mechanism, articulated originally in Pedro et al. (2018), is at least partly responsible for the slow rebound in the initial wind-driven cooling at depth near the AIS.

## 4 Discussion

Peak warming during the LIG is believed to have occurred after the 4–5ka long Heinrich-11 event, though uncertainty remains in the length and timing of this event. Proxy data indicate global LIG warming of about 1 °C with respect to PI (Dutton et al., 2015), and about 1.6 ± 0.9 °C at high latitudes relative to present day (Turney et al., 2020; Capron et al., 2017; Hoffman et al., 2017). The CESM2-FV2 *127ka* run simulates global average air temperatures only marginally warmer (0.004°C) than the *piControl* (Fig. A2a). That climate models tend to estimate weaker LIG warming than proxies by about 1°C is typical

(Turney et al., 2020). Southern Ocean SST's in CESM2 are warmer by about 0.1°C, and over the Antarctic continent, annual near-surface air temperatures are warmer than PI by about 0.4°C (Table 2).

With additional freshwater forcing in the North Atlantic in the *127kaFW* simulation, a strong bipolar response is established within a few centuries and persists through the simulations. When compared to the available LIG proxy evidence (Fig. A2c), the model's bipolar seesaw air temperature response is exaggerated. This is particularly true in the NH, where modeled tem-

peratures are colder than proxies by more than 5°C (Fig. A2d). Similarly, three of the four ice core locations in the Southern





Hemisphere are warmer in the model than the cores, while many Southern Ocean locations are warmer in the model than the proxy by more than 4°C.

We note that while the 1° (CESM2-FV1) *127ka* simulation produces a larger global temperature anomaly (0.11°C) (Otto-Bliesner et al., 2020) than our 2° simulation (CESM2-FV2), our 2° run generates the same regional and seasonal patterns.

Positive temperature anomalies are strongest in JJA over the Northern Hemisphere continents (upwards of 4°C warmer than PI in large swaths of North America and Asia), while the strongest negative DJF anomalies occur over almost all large land features except Greenland. During *127ka*, DJF global air temperatures are colder than PI by roughly 1°C, and JJA temperatures are warmer by slightly more than 1°C (Fig. A5). In DJF, the Arctic and parts of the Southern Ocean warm.

Changes in air temperature of even a few degrees are unlikely to significantly impact ice sheet mass balance, given the

already cold continental climate. Instead, the ice–ocean interface of Antarctica was probably the primary driver of Antarctic mass loss in the past, as it is today (Noble et al., 2020).

Subsurface warming can reach up to 0.4°C at some locations near the ice sheet in the *127ka* simulation compared to *piControl*. This magnitude of change, though modest, could result in significant ice mass loss if persistent over long periods of time. In their ice sheet model experiments, for example, Turney et al. (2020) achieve WAIS collapse (4.5m sea level equivalent mass

loss) under only 1°C ocean temperature forcing within two millennia. Berdahl et al. (2023) found that under ocean conditions comparable to today in the Amundsen sector, ice shelves lifted off key pinning points and unfettered retreat was initiated within five centuries. Given that today's Amundsen sector is likely warmer than PI by a few tenths of a degree, we argue that the modest CESM2 warming seen in the *lig127ka* run provides a reasonable estimate of actual LIG warming.

Other recent work suggests that meltwater discharge from the AIS could amplify mass loss by stratifying the ocean column

and trapping upwelling warm water beneath ice shelves, increasing the thermal forcing at the ice–ocean interface (Golledge et al., 2019). We speculate that given the non-linear response following loss of buttressing, and the possibility of positive feedbacks due to local meltwater discharge, a small but persistent thermal forcing anomaly of ~0.4°C (as generated in the *127ka* simulation) could cause significant mass loss over several millennia.

As noted, the PI mean state in CESM2 is known to be too warm in the subsurface Southern Ocean (A1). Also, the ocean

model has a coarse 1-degree resolution, and does not represent sub-shelf cavities. Therefore, there remains plenty of uncertainty in whether the *127ka* run simulates the correct magnitude of warming and whether it warms for the right reasons.

Instead of the freshwater forcing in *127kaFW* amplifying Southern Ocean warming, we found that the *127kaFW* simulation shows unexpected subsurface cooling in the Southern Ocean near the AIS. This cooling is strongest in the first millennium, especially in high southern latitudes along the Antarctic coastline. Subsurface temperatures then rebound slowly, taking millen-

nia to reach original values, and in some cases not reaching original temperatures by the end of the 4000-year run. Subsurface ocean cooling at high southern latitudes in response to North Atlantic freshwater fluxes occurs in other model runs, but has not been identified as a noteworthy dynamical response in previous work. However, it may be a robust feature of the system. For example, the CCSM4 iTrace (isotope-enabled) simulation (He et al., 2021) includes evolving freshwater fluxes in the North Atlantic that eventually spread to the Southern Ocean. This is meant to simulate the last deglaciation (Heinrich 1) from the last

glacial maximum (LGM, ~20ka) to the early Holocene (11ka). We find that subsurface temperatures (upper 1000 m) in the



TightSO cool slightly ($\sim 0.1^\circ$C) but persistently, lasting about a millennium, in response to the initial Atlantic freshwater fluxes in the simulation (Heinrich Stadial 1) (Fig. A7). This result is also seen in Zhu et al. (2022), who showed subsurface cooling (at 1 km depth) south of the ACC in response to the Heinrich 1 event (see their Fig 5). In another PMIP4 Tier 1 *lig127k-H11* simulation, which follows the same forcings prescribed in our experiments and in Otto-Bliesner et al. (2020), Guarino et al.

(2023) observed cooling at a depth of 500 m in high southern latitudes following the freshwater forcing. They attributed these changes to increases in Antarctic sea ice area, but acknowledged that the sea ice increase may be a result of the limited length of their simulations (250 years). In contrast, our freshwater experiment shows a decrease in Antarctic sea ice, resulting in about a $\sim 30\%$ loss, consistent with warming SSTs in the Southern Ocean. This reduction is consistent with the two-timescale sea-ice response proposed by Ferreira et al. (2015), and with results from HadCM3 simulations of the last interglacial showing

Antarctic sea ice reduction (Holloway et al., 2018). Sea ice proxy records for 128ka show a significant reduction in winter sea ice extent (Holloway et al., 2017), consistent with both our *127ka* and *127kaFW* simulations.

The initial subsurface ocean cooling in our *127kaFW* simulation is evidently related to wind-driven changes in ocean circulation as a result of freshwater flux in the North Atlantic. As peak westerlies shift southward, upwelling close to the continent increases, moving cold water up through the column and bringing colder ocean temperatures to depths relevant to the ice sheet.

This is followed by a slow rebound in subsurface temperatures as heat stored in the Southern Ocean north of the ACC is transported via eddies across this dynamic barrier. A comprehensive discussion of this mechanism of eddy propagation across the ACC can be found in Pedro et al. (2018).

What are the implications for this subsurface cooling in terms of WAIS collapse during the Last Interglacial? Based on our CESM2-FV2 simulations, the subsurface ocean conditions near the AIS are more favorable to melting at the ice–ocean

interface in the *127ka* simulation than in the *127kaFW*. While the actual freshwater forcing from Heinrich 11 was probably a significant driver of warming ocean temperatures in the real world (Clark et al., 2020), our study shows that, depending on timing and duration, freshwater forcing need not necessarily lead to greater thermal forcing of the ice sheet margin. Given that we do not expect a large freshwater forcing of comparable scale to the H11 event in the future, our *127ka* simulation – without freshwater forcing – may be a more relevant analog for the future. Notably, although the global mean temperature response

in this simulation is negligible, warming in the key Amundsen Sector occurs at depths that are relevant to the ice sheet. This suggests that the possibility of WAIS collapse during the LIG, under conditions without freshwater forcing, should be further explored.

Finally, we emphasize that the connection between ice mass loss and ocean thermal forcing is complex. The factors that determine how heat is moved onto the continental shelf are not well observed and operate on many temporal and spatial scales

(e.g., topography and slope of the continental shelf, eddy mixing, waves, presence and location of the Antarctic Slope Front, and proximity to CDW) (Noble et al., 2020). Once heat is on the continental shelf, it is a combination of processes that determine melt rates, including sub-shelf cavity circulation, bathymetry, tides (Jourdain et al., 2019) and millimeter-scale turbulence and convection at the ice-ocean boundary (Noble et al., 2020). Studies using high-resolution ocean modelling to capture sub-shelf circulation are emerging (e.g., Nakayama et al. (2018)) and hold promise for developing a more comprehensive picture of the

relationship between offshore ocean conditions and basal melt rates of the ice sheet.



## 5   Conclusions

This study analyzed two CESM2-FV2 paleoclimate simulations: one configured with Last Interglacial orbital and greenhouse forcing, and the second with additional freshwater forcing in the North Atlantic. We examined the global climate response with respect to the pre-industrial climate, with particular attention to climate conditions near the Antarctic ice sheet.

While proxy evidence suggests global temperatures were about 0.5–1°C warmer during the LIG (Arias et al., 2021; Dutton et al., 2015), the *127ka* run simulates only ∼0.004°C warming compared to the *piControl*. Similarly, Antarctic air temperatures increase less than suggested by proxy evidence, but are within the margin of error in the proxy records. Regional and seasonal patterns of warming (e.g., warm summers over NH continents) are consistent with higher-resolution models (e.g., Otto-Bliesner et al. (2021)).

Despite the negligible change in global mean temperature, ocean temperatures in *127ka* increase by up to 0.4°C compared to the pre-industrial at depths relevant to the margin of the Antarctic ice sheet. This increase in thermal forcing appears to be a robust Southern Ocean response to the insolation changes expected under LIG conditions. If applied to the grounding line at some key ice shelves for long enough, it may have melting potential. However, we cannot assume that 0.4°C warming off the shelf translates to 0.4°C sub-shelf warming. Rather, processes that transport warm water to the ice shelf are crucial to
determining thermal forcing at the ice-ocean interface.

As expected, the addition of freshwater in *127kaFW* causes AMOC collapse, reduction of northward Atlantic heat transport, and a resulting bipolar temperature response with a cooling NH and warming SH at the surface. Antarctic sea ice declines within the first few centuries of the simulation, followed by a slower but steady decline throughout the rest of the simulation. Ocean heat content increases in all major ocean basins. However, in the high southern latitudes (TightSO), heat content decreases after
the freshwater forcing begins. The atmospheric circulation shifts swiftly in response to the hosing and AMOC collapse. In the Southern Ocean, a poleward shift in peak westerlies remains shifted by several degrees latitude for the rest of the simulation. The surface wind anomalies lead to changes in wind stress curl and force changes in local ocean circulation. Negative wind stress curl anomalies near the AIS suggest deflection of surface waters to the left allowing upwelling of colder subsurface water. The wind-driven subsurface cooling lasts for the first millennium of the simulation, after which temperatures rebound slowly
but never recover to their original values in many locations around the AIS. The slow temperature rebound may be related to eddy propagation of heat across the ACC, supported by strengthening of the eddy component of the Southern MOC cell.

There are implications for the thermal forcing (and thereby mass balance) of the AIS in this pair of simulations. Primarily, the orbital-only forced simulation generates more potential ocean thermal forcing at depths relevant to Antarctic ice sheet melt than the simulation that includes freshwater forcing. This was an unexpected result, but appears to be a non-unique model
response to freshwater forcing in the North Atlantic.

Based on these simulations, we suggest that the *127ka* run is a more relevant analog to future climate than either the *127kaFW* run or other simulations that include large freshwater fluxes in the North Atlantic. Future work includes forcing standalone ice sheet models with the thermal forcing anomalies from these simulations. This will allow a more complete assessment of the combined ocean–ice conditions of WAIS collapse. Another priority will be to work toward more realistic simulations of the

Southern Ocean in future versions of CESM. Reducing the mean state biases in the PI simulations would reassure us that we are capturing the direction and magnitude of LIG anomalies. Finally, high-resolution models ocean models are needed to link off-shelf warming to sub-shelf melting.

*Code availability.* CESM is an open-source code developed on the Earth System Community Model Portal (ESCOMP) git repository at *https://github.com/ESCOMP/CESM*. The CESM version used in this work was tagged as release-cesm2.1.1 and is available at *https://github.com/ESCOMP/CESM/tree/release-cesm2.1.1*.

*Data availability.* The data and configuration files will be made publicly available on the Research Data Archive (*https://rda.ucar.edu/*) and will be advertised on the NSF NCAR Paleo Working Group simulation webpage (*https://www.cesm.ucar.edu/working-groups/paleo/simulations*).

*Author contributions.* MB wrote the manuscript with contributions from all authors. GL carried out the *127ka-FW* experiment extension beyond 1000 years. RT carried out the *127ka* and the first 1000 years of the *127ka-FW* experiment. MB analysed the data. ES, WL and BO-B supervised the project. All authors discussed the results and helped shape the research, analysis and manuscript.

*Competing interests.* The authors declare no competing interests.

*Acknowledgements.* This study was supported by National Science Foundation grant no. 2045075. Gunter Leguy, William Lipscomb, and Bette Otto-Bliesner were supported by the NSF National Center for Atmospheric Research, which is a major facility sponsored by the National Science Foundation under Cooperative Agreement no. 1852977. Computing and data storage resources for CISM simulations, including the Cheyenne supercomputer (https://doi.org/10.5065/D6RX99HX), were provided by the Computational and Information Systems Laboratory (CISL) at NSF NCAR.





**Appendix A**

**Climate**
**of the Past**
Discussions

EGU



**Figure A1.** Comparison of modeled (*piControl* and *127ka*) and observed (Estimating the Circulation & Climate of the Ocean (ECCO)) ocean temperature (top row) and salinity (bottom) profiles at three locations off the Amundsen Sea Coast. CESM2 has a (known) warm bias below the thermocline. Note: The y axis range is different at each location (depending if it is on or off the continental shelf).

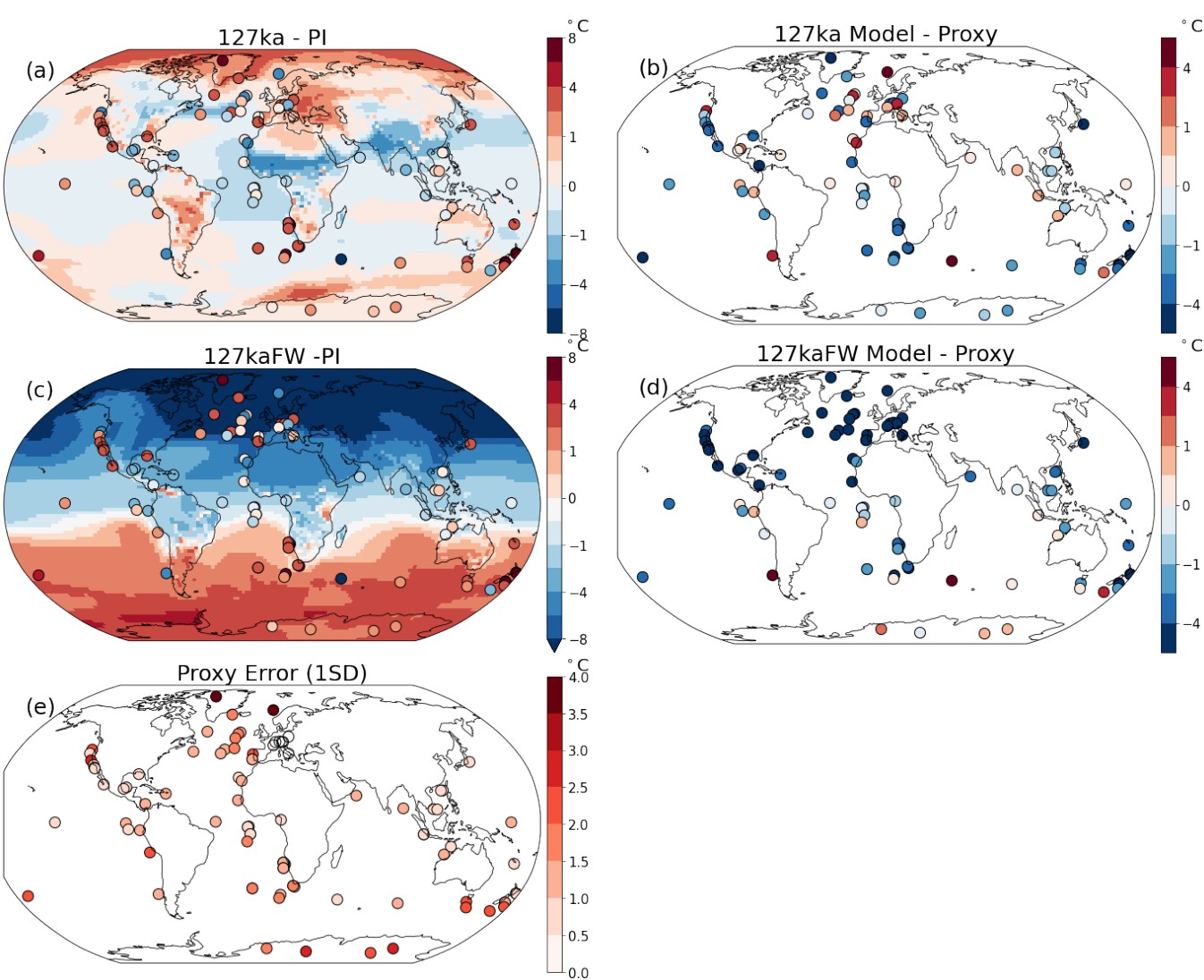

**Figure A2.** Climatological 2m air temperature anomaly with respect to *piControl*. *127ka* (a) and *127kaFW* (c). Climatologies are computed over the last 100 years of the simulations. Proxy data indicating anomalies between 127ka and PI are overlain (data from Otto-Bliesner et al. (2020); Capron et al. (2017). Difference between model climatology and proxy for *127ka* (b) and for *127kaFW* (d). Errors associated with proxies (1 standard deviation) (e). NB: Antarctic proxies scaled as in Fig. 2.



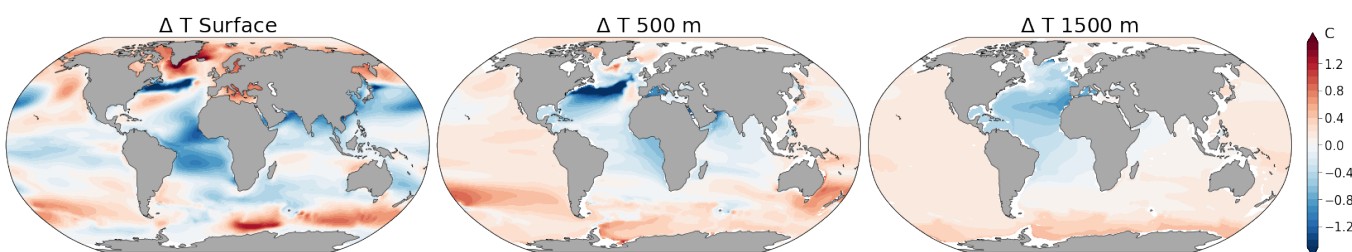

**Figure A3.** *127ka* ocean temperature anomaly with respect to *piControl* for ocean surface (left), 500 m depth (center) and 1500 m depth (right).



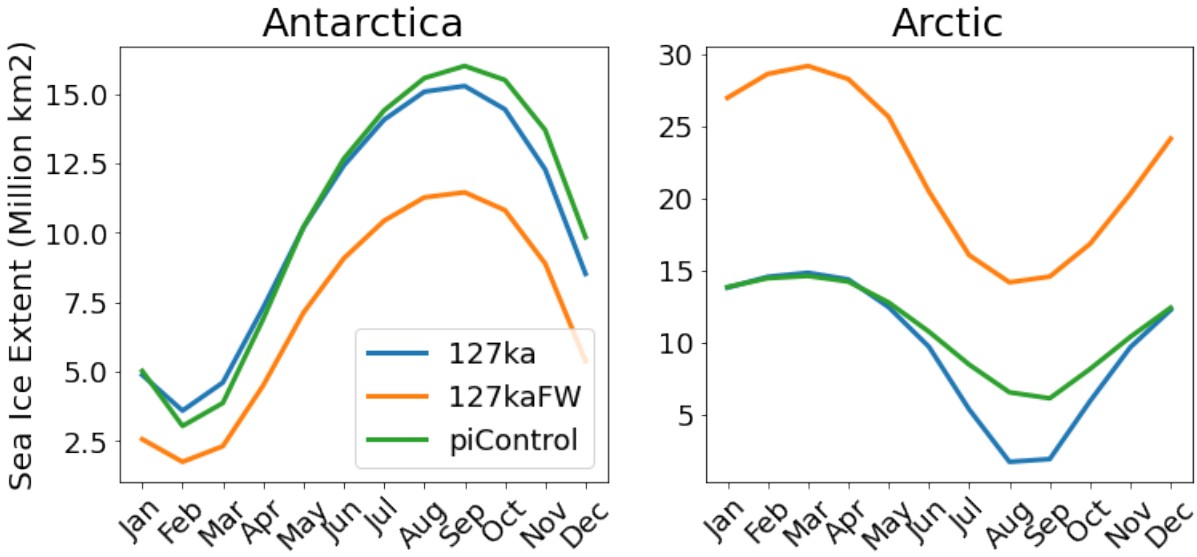

**Figure A4.** Seasonal sea ice extent for the Antarctic and Arctic. Seasonal climatology for all three simulations are taken over the last 50 years of the simulation.




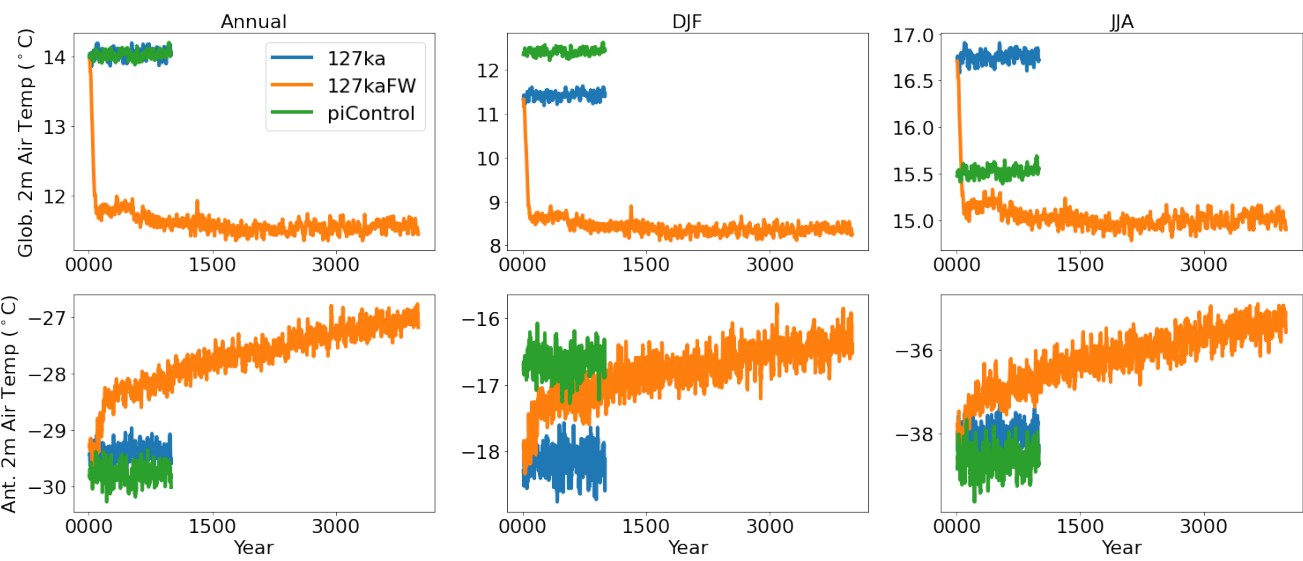

**Figure A5.** Global (top row) and Antarctic (bottom row) 10-year running mean of annual (left), DJF (center) and JJA (right) 2m air temperature. Colors indicate different simulations.





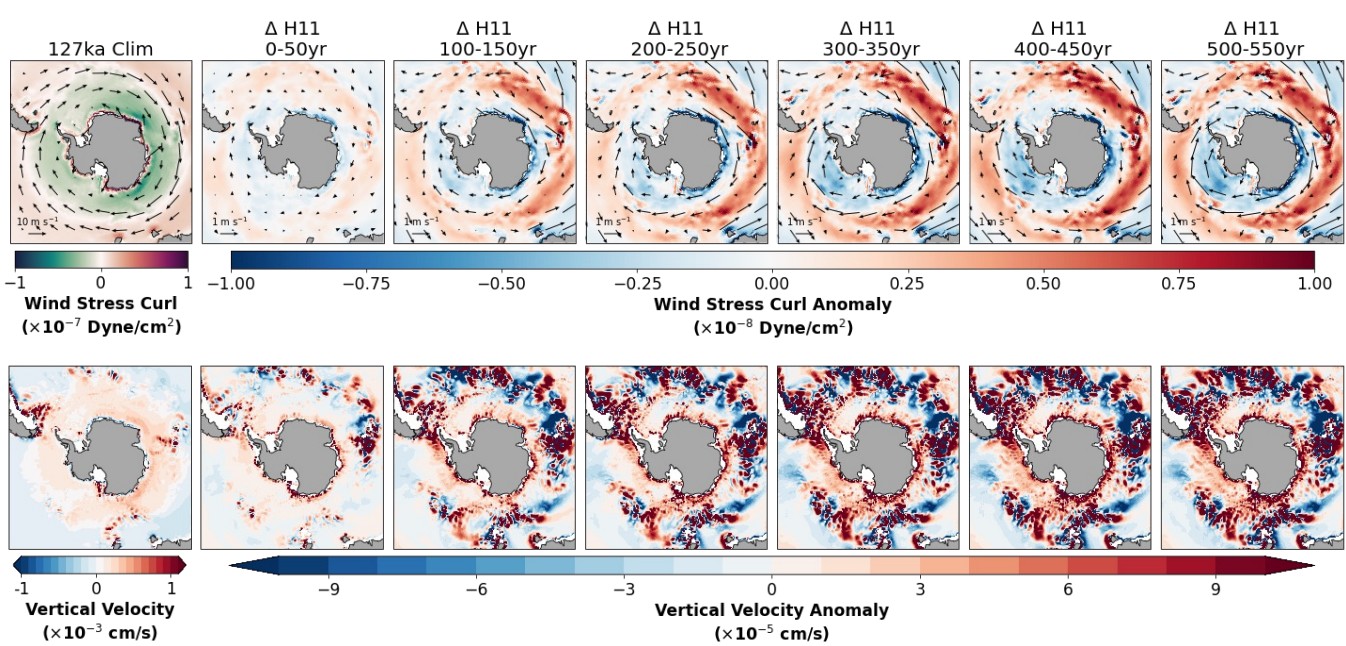

**Figure A6.** Surface winds and wind stress curl (top row), and vertical ocean velocity at 250m depth (bottom row) over the first 500 years of the *127kaFW* simulation. Left panels show *127ka* climatology, while subsequent panels in each row indicate anomalies between *127kaFW* and *127ka*. *127kaFW* snapshots are taken as 50 year means, as indicated in top labels.





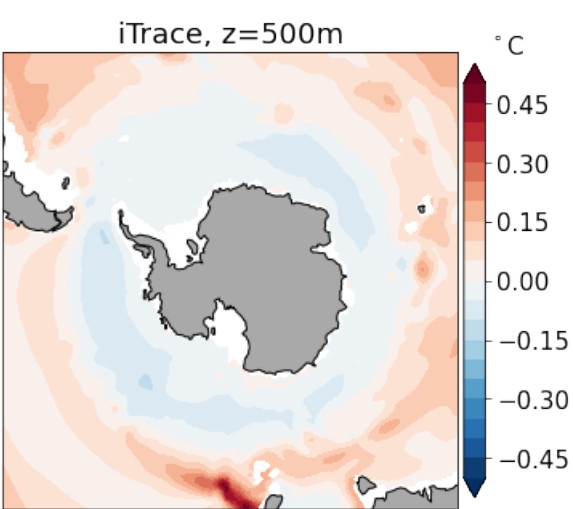

**Figure A7.** iTrace ocean temperature anomaly at 500m depth between 50-year climatology at years 950–1000 compared to 50-year climatology at the simulation start. This timeframe is chosen because it is the most analogous simulation period to our *127kaFW* simulation. That is, it is the only period that is purely influenced by North Atlantic freshwater forcing. The rest of the iTrace run includes other forcings that complicate the response. For more on iTrace, see He et al. (2021).



|  | Southern Ocean | | | Tight Southern Ocean | | | Amundsen | | | Tight Amundsen | | |
|---|---|---|---|---|---|---|---|---|---|---|---|---|
|  | DJF | JJA | Ann | DJF | JJA | Ann | DJF | JJA | Ann | DJF | JJA | Ann |
| SST | -0.02 | 0.12 | 0.1 | -0.15 | 0 | -0.05 | -0.05 | 0.16 | 0.11 | -0.46 | -0.02 | -0.19 |
| 200m | 0.26 | 0.24 | 0.25 | 0.15 | 0.14 | 0.15 | 0.41 | 0.39 | 0.40 | 0.24 | 0.23 | 0.23 |
| 500m | 0.22 | 0.22 | 0.22 | 0.25 | 0.24 | 0.24 | 0.3 | 0.3 | 0.29 | 0.19 | 0.19 | 0.19 |
| 750m | 0.17 | 0.17 | 0.17 | 0.24 | 0.24 | 0.24 | 0.2 | 0.2 | 0.19 | 0.18 | 0.18 | 0.18 |
| 1000m | 0.16 | 0.16 | 0.16 | 0.26 | 0.26 | 0.26 | 0.16 | 0.16 | 0.16 | 0.19 | 0.19 | 0.19 |
| 1500m | 0.16 | 0.16 | 0.16 | 0.27 | 0.27 | 0.27 | 0.15 | 0.15 | 0.15 | 0.21 | 0.21 | 0.21 |

|  | Tight AP | | | Tight Ross | | | Tight Weddell | | | Tight EAIS | | |
|---|---|---|---|---|---|---|---|---|---|---|---|---|
|  | DJF | JJA | Ann | DJF | JJA | Ann | DJF | JJA | Ann | DJF | JJA | Ann |
| SST | -0.17 | 0 | -0.06 | -0.22 | 0 | -0.07 | -0.12 | 0 | -0.03 | 0.28 | 0.02 | 0.13 |
| 200m | 0.17 | 0.16 | 0.16 | 0.17 | 0.15 | 0.16 | 0.14 | 0.13 | 0.14 | 0.01 | 0.02 | 0.02 |
| 500m | 0.23 | 0.23 | 0.23 | 0.19 | 0.18 | 0.19 | 0.38 | 0.38 | 0.38 | 0.24 | 0.23 | 0.24 |
| 750m | 0.2 | 0.2 | 0.2 | 0.22 | 0.22 | 0.22 | 0.34 | 0.34 | 0.34 | 0.24 | 0.24 | 0.24 |
| 1000m | 0.16 | 0.16 | 0.16 | 0.26 | 0.26 | 0.26 | 0.31 | 0.3 | 0.31 | 0.26 | 0.26 | 0.26 |
| 1500m | 0.14 | 0.14 | 0.14 | 0.29 | 0.29 | 0.29 | 0.3 | 0.3 | 0.3 | 0.26 | 0.26 | 0.26 |

**Table A1.** Regional and seasonal climatological ocean temperature differences, *127ka - piControl* (°C). Climatologies are defined as final 50 years of each run.



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
