# Peer review of "Antarctic climate response in Last-Interglacial simulations using the Community Earth System Model (CESM2)"

_Climate of the Past, 2024_

## Referee Comment (RC1)

**Antarctic climate response in Last-Interglacial simulations using the Community Earth System Model (CESM2)**

by Berdahl M., Leguy G. R., et al.

**General Comments**

In this study the authors present CESM2 simulations of the Last Interglacial climate following the CMIP6-PMIP4 protocol. Compared to previously published CESM2 PMIP4 simulations, they use here a 2degrees nominal resolution for the atmosphere and land models and replace the PI vegetation (as per protocol) with a 'potential vegetation' considered to be more representative of the LIG climate since urban areas are not present. All other boundary and initial conditions as per protocol.

Authors present results from a LIG climate equilibrium simulation (at 127ka) and a LIG climate freshwater experiment (FW). A preindustrial (PI) simulation is also discussed. They focus on analysing the Southern Ocean and Antarctic response to the LIG and freshwater forcing. Particularly, the authors draw the reader's attention to a possibly overlooked climate system response to the freshening of the North Atlantic: a cooling in the subsurface ocean near the Antarctic ice sheet.

Generally, I found the manuscript clearly written, motivated and with a good introduction to the topic. The length of the simulations here presented (1000 years for PI and LIG, and 4000 years for the FW experiment) is certainly a point of strength of this study, given that fully coupled climate simulations this long are not frequent.

I have a number of (overall) minor comments that I ask the authors to address before recommending the manuscript for publication.

The feeling I had while reading the paper is that often differences between the model runs are described without providing a thorough physical explanation of why they exist in the first place. Particularly, I refer to the warming of the Southern Ocean, at depths, for the LIG experiment and to the subsurface cooling of the Antarctic waters for the FW experiment (which is also the main result of this study).

**Specific Comments**

Line 12: is 'new' the right word? For example, Guarino et al. 2023 showed this too for PMIP4 HadGEM3 simulations (see their Figure 3 and Suppl.Fig.1).

Lines 56-75: you mean similar runs using CESM2? the literature is quite vast for freshwater experiments under LIG conditions using different models.

Line 66: I don't understand this sentence: "in anticipation of running coupled experiments in future work". These experiments are already coupled, aren't they?

Section 2.1: the length of these simulations is a point of strength of this paper. However, what is the model spin-up? (see also my comment below).

Line 112: why using only 50 years when you have 1000? 50 years will not be enough to sample the climate variability of processes that manifest on longer timescales (e.g. centennial climate oscillations). Is the spin-up period included in these 1000 years? this is not mentioned in methods.

Lines 112-113: Please add more details about the analysis presented in section 3.2, as it was done for 3.1. For example, how are anomalies computed? (FW-LIG?), how many years of simulation did you use for your analysis?

Lines 130-132: can you show the increase in the westerlies in your 2degree simulation too? In Otto-Bliesner et al., 2020 this point is indeed made but I could not find a figure showing the stronger westerlies (I have not checked if there are SI figures, but there is no mention of this in their paper).

Line 133: "(..more vigorous transport, not shown)"
I suggest you include this in your Appendix, this is a rather crucial aspect for your following analysis of the SO (see also my comment about this at line 189).

Line 134: where is this shown?

Lines 168-169: important point, if robust, add to conclusions(?)

Line 170: is the regional cooling in these sectors caused by sea ice changes?

Line 189: SST warm anomalies can be linked to the LIG insolation forcing, but what is the dynamical link between LIG forcing and SO warming throughout the water column?
Is it maybe the increased global oceanic heat transport under LIG forcing discussed in Otto-Bliesner et al. (2020) that was mentioned earlier? Is the increase in ocean heat transport for the LIG versus the PI shown in Otto-Bliesner et al. (2020)?

General comment on section 3.1.2: up to this point in the paper changes between mean state PI and LIG are described but not discussed, why are we seeing these changes? what is driving them? why is the ocean warmer at depth during the LIG? why are the westerlies weaker?

Figure 4, caption: "Climatologies are computed over the last 50 years of each simulation."
You can omit this, as it is in your methods.

Figure 5, panel e: I do notice at the very beginning of your 127kaFW-SH timeseries a small bump, suggesting SH sea ice initially increased before starting declining. Have you looked into this? (I am asking out of personal curiosity).
It is difficult for me to appreciate from this graph how many years it took for the sea ice to begin melting. This might be the transient sea ice response that we discussed in Guarino et al., 2023, and the subsequent decline in your simulation might confirm the two time-scales response of sea ice first proposed by Ferreira et al. 2015 and then invoked by Guarino et al. 2023 to explain the increase of Antarctic sea ice under H11 forcing.

Line 225: should "127ka" be 127kaFW?

Figure 7, panel a: "over the Antarctic continent"
How is this defined? land points only?

Figure 8: This figure is key to the paper results, and I would like to understand it better. In particular, I would like the authors to better substantiate some of their analysis. There will be some repetition between my remarks here and some comments below.

If we look at the Weddell and the Ross/Amundsen Sea sectors, they behave in opposite ways, and I feel this aspect is not sufficiently discussed in the manuscript.

Even more important, I don't understand why the authors talk about upwelling of cold waters (line 359). Generally, an increase in surface winds (and particularly in the negative wind curl in the SH) means, yes, increased upwelling (because the flow is divergent at the surface) but of relatively warm waters. This is a well-known mechanism for triggering open ocean deep convection and polynyas formation in CMIP models in the Weddell Sea.

In fact, if I had to guess, I would think that the weaker (with time) winds and wind stress over the Weddell weaken the Weddell Gyre and discourage the occurrence of open ocean deep convection in the FW experiment compared to the LIG. Less Deep convection could imply less warm water entering the 200-800m water column, and thus the negative MOT anomaly persists in the Weddell.

The same is not true for the Ross Sea.
In the Ross, as I also say in my comment at line 281-283, I can't really appreciate a strong trend in wind stress curl, but if there is one (as I think the authors say) then I would expect the same mechanism as above to be active: upwelling of warm waters as the Ross gyre spins up. Some models (not all) do simulate deep convection also in the Ross Sea (see de Lavergne et al., NCC, 2014) although I don't know where CESM stands on this.

Is it possible that the upwelling of warm waters in your bottom panels is linked to this?

Finally, I believe the establishment of the initial MOT negative anomaly needs explaining.

Line 264: "Within the first 500 years, a bulwark of cold subsurface water wrapping around the continent"
What is driving this?

Lines 280-281: is this shown? does this mean that the SAM remains more positive than the LIG during the whole LIG-FW experiment?

Lines 281-283: Okay the physical mechanism, but in Fig.8 the wind stress weakens with time, particularly in the Weddell. As for other regions, Ross and AS, if there is a trend (i.e. wind curl becoming more and more negative) this

cannot be appreciated from Figure 8 where I see almost no difference from 500 to 4000yrs.

Lines 286-287: I feel that the reasons for the subsurface cooling in the SH are not well explained anywhere in the manuscript, if this has been shown in other studies references plus a short description of the physical mechanisms at play must be included.

Line 291: has 'CDW' been defined before?

Line 338: Is this truly unexpected? There is some literature about this.

Guarino et al.,2023 (as you mention later on) and some of the references in there provided:

Crowley, T. J. and Parkinson, C. L.: Late Pleistocene variations in Antarctic sea ice II: effect of interhemispheric deep-ocean heat exchange, Clim. Dynam., 3, 93–103, 1988.

Renssen, H., Goosse, H., Crosta, X., and Roche, D. M.: Early Holocene Laurentide Ice Sheet deglaciation causes cool- ing in the high-latitude Southern Hemisphere through oceanic teleconnection, Paleoceanography, 25, PA3204, https://doi.org/10.1029/2009PA001854, 2010.

Lines 341-342: I agree on this.

Lines 353-355: Well, this point exactly is the one we make in Guarino et al. 2023, in which the short timescale response is analysed and where HadCM3 results are invoked to explain the long time scale response.
In fact, it would be quite interesting to take a look at your first 200-300 years of simulations and see if results are consistent with HadGEM3 ones, confirming the transient sea ice response.

Line 370: Again, I feel this dynamical response of the system has been well explained. Why does this happen in the LIG experiment?

Lines 385-389: This paragraph seems a repetition, the Discussion section begins in the same way. Consider keeping only one of the two.

*Maria Vittoria Guarino*

---

## Referee Comment (RC2)

**Review of "Antarctic climate response in Last-Interglacial simulations using the Community Earth System Model (CESM2)"**

by Mira Berdahl, Gunter R. Leguy, William H. Lipscomb, Bette L. Otto-Bliesner, Esther C. Brady, Robert A. Tomas, Nathan M. Urban, Ian Miller, Harriet Morgan, and Eric J. Steig

The climate of the Last Interglacial was characterized by warmer temperature and higher sea level, in response to different orbital parameters. The sea level rise was likely partly due to a partial West Antarctic ice sheet melt, possibly due to Southern Ocean warming. The ocean warming could be related to an AMOC weakening induced by fresh water originating from Northern ice sheet melting. To better understand what could have caused an Antarctic ice sheet melt, Berdahl et al. analyse two CESM2 simulations run with 127ka orbital parameters, including one with fresh water fluxes. The simulations are compared to data with a focus on Antarctica to evaluate the possibility of favouring an Antarctic ice sheet melt.

Neither of the two simulations capture the full magnitude of the temperature increase at 127ka recorded in proxy data, but the 127ka simulation shows a small ocean temperature increase that could favour West Antarctic melt. The 127ka simulation with fresh water fluxes displays a local cooling around Antarctica despite general warming in the Southern Hemisphere, indicating that this is unlikely to explain the Antarctic ice sheet melt according to CESM2.

The paper is well written and well organized, and the two simulations are interesting. But this study is relatively limited. It would have been more comprehensive and useful to test the impacts of Antarctic changes such as local fresh water fluxes and a smaller ice sheet to evaluate the impact on climate and possible feedbacks.

**General comments**

Since the focus is the Antarctic climate and the possibility of the Antarctic ice sheet collapse it would have been interesting to test the impact of fresh water flux from Antarctic, whether this could trigger a positive feedback. There is a small discussion on it, this could be developed.

On the same topic, it would have been great to test the impact on climate of having a smaller Antarctic ice sheet and the possible feedback through ocean temperature and surface mass balance changes.

Finally, the fresh water flux addition results in large changes in disagreement with proxy data, but a smaller fresh water flux could improve the model-data comparison.

Would it be doable to do one (or several) additional simulations? How long does it take to run 1000 years? Additional simulations could be shorter than the ones presented here, even if the equilibrium is not reached it would still be interesting.

Three complementary simulations might be worth considering:

-   With 127ka ice sheets / smaller West Antarctic ice sheet (this could have a regional impact)

- With fresh water fluxes in the Southern Ocean
- With less intense fresh water fluxes in the North Hemisphere

and possibly a combination of the three.

Since the number of figures is reasonable, and looking at the additional figures while reading the manuscript is not very comfortable, I would advise to switch some of the additional figures to the main text, especially those including Southern Ocean temperature (as this is the most relevant for possible Antarctic ice sheet melt) such as figures A2, A3 and A5.

**Specific comments**

Abstract

Could you include a sentence on the comparison with data and which simulation is best, or more likely?

Model description
Line 67 explain what is FV1? 1-degree model? It is mentioned as the 1-degree model but only much later (p.19).

How long are the simulations? What is the gain compared to the 1degree resolution version?

Results
Line123: "the orbital-only forcing underestimates" -> the orbital-only simulation underestimates
Line 125-126: can you give the temperature change in CESM 1degree to compare with? And also, you could compare to other models as described in Otto-Bliesner 2021.

Line134-135: are you showing this (AMOC change) somewhere? Or are you referring to another paper? Is the AMOC change the same in the two model versions?

Figure 3. What is the variable plotted in the background? Either specify the variable adding units and colorbar if it is useful to keep, or remove the background colors to only keep the different regions.

Section 3.2.2
The eddy component seems to play an important role. Can you elaborate on how eddies are represented and how this could change (or not) in a higher resolution model? In other model experiments with fresh water fluxes (apart from CESM, and in other than LIG period), has this cooling been observed before? Is this robust or model dependant?

Discussion

Line 307-312. The bipolar seesaw response is too large with the imposed fresh water flux. Could you discuss the possibility of having a better response with a smaller fresh water? It would be interesting to have a simulation with a smaller fresh water flux to evaluate the climate response.

Line 329- 333. It would have been great to have a simulation with a fresh water flux in the Southern Ocean to evaluate its impact.

Line 337- 342. You could discuss the implication in terms of transient evolution if this cooling and delayed warming is a robust feature: could it result in preventing the Antarctic ice sheet from melting too early when the Northern ice sheets are melting, and allow the Antarctic ice sheet to melt later?

Conclusions
Do you have ideas on what could be missing to obtain temperature changes in better agreement with proxy data?

---

## Author Comment (AC1)

**Reviewer 1**

Antarctic climate response in Last-Interglacial simulations using the Community Earth System Model (CESM2) by Berdahl M., Leguy G. R., et al.

**General Comments**

In this study the authors present CESM2 simulations of the Last Interglacial climate following the CMIP6-PMIP4 protocol. Compared to previously published CESM2 PMIP4 simulations, they use here a 2degrees nominal resolution for the atmosphere and land models and replace the PI vegetation (as per protocol) with a 'potential vegetation' considered to be more representative of the LIG climate since urban areas are not present. All other boundary and initial conditions as per protocol.

Authors present results from a LIG climate equilibrium simulation (at 127ka) and a LIG climate freshwater experiment (FW). A preindustrial (PI) simulation is also discussed. They focus on analysing the Southern Ocean and Antarctic response to the LIG and freshwater forcing. Particularly, the authors draw the reader's attention to a possibly overlooked climate system response to the freshening of the North Atlantic: a cooling in the subsurface ocean near the Antarctic ice sheet.

Generally, I found the manuscript clearly written, motivated and with a good introduction to the topic. The length of the simulations here presented (1000 years for PI and LIG, and 4000 years for the FW experiment) is certainly a point of strength of this study, given that fully coupled climate simulations this long are not frequent.

I have a number of (overall) minor comments that I ask the authors to address before recommending the manuscript for publication.

The feeling I had while reading the paper is that often differences between the model runs are described without providing a thorough physical explanation of why they exist in the first place. Particularly, I refer to the warming of the Southern Ocean, at depths, for the LIG experiment and to the subsurface cooling of the Antarctic waters for the FW experiment (which is also the main result of this study).

We thank the reviewer for their thorough and thoughtful comments. We have tried to address all their comments, especially those concerning mechanisms driving subsurface ocean warming in 127ka and the initial cooling in 127kaFW. We hope this clarifies the drivers behind these changes.

**Specific Comments**

Line 12: is 'new' the right word? For example, Guarino et al. 2023 showed this too for PMIP4 HadGEM3 simulations (see their Figure 3 and Suppl.Fig.1).

We have removed "new". Yes, others have seen this behavior, we only meant to suggest that the behavior has not been explicitly discussed, nor explained.

Lines 56-75: you mean similar runs using CESM2? the literature is quite vast for freshwater experiments under LIG conditions using different models.

This is true. The text now specifies that we mean this for similar CESM runs in particular. "Previous work evaluating the large-scale features of similar CESM LIG runs (e.g., Otto-Bliesner et al., 2020, 2021) showed that the large positive NH solar insolation anomaly results in summer warming over the NH continents and reduced Arctic summer minimum sea ice."

Line 66: I don't understand this sentence: "in anticipation of running coupled experiments in future work". These experiments are already coupled, aren't they?

We meant to say fully coupled with a dynamic Antarctic ice sheet.
The text has now been updated to read:
"We use the relatively low resolution of FV2 in anticipation of running coupled experiments with a dynamic Antarctic ice sheet in future work, to save on computational expense."

Section 2.1: the length of these simulations is a point of strength of this paper. However, what is the model spin-up? (see also my comment below).

The 2-degree PI Control run is spun-up for 300 years. The 2-degree 127ka Control run is spun-up for 450 years under PI conditions, followed by 460 years under orbital changes. The 127kaH11 is branched from the 127ka Control run, so it has the same spin-up as the 127ka Control run. This information has been added to Table 1.

Line 112: why using only 50 years when you have 1000? 50 years will not be enough to sample the climate variability of processes that manifest on longer timescales (e.g. centennial climate oscillations). Is the spin-up period included in these 1000 years? this is not mentioned in methods.

The spin-up is not included in these 1000 year production runs. This is now made clear in the methods section with additional text: "All simulations are spun-up before the millennial integrations take place, and spin-ups are not included in the analyses."
We use only 50 years when computing climatological means since there is little model drift in the control runs.

Lines 112-113: Please add more details about the analysis presented in section 3.2, as it was done for 3.1. For example, how are anomalies computed? (FWLIG?), how many years of simulation did you use for your analysis?

At the beginning of Section 3.2 we state that all anomalies in 127kaFW are calculated with respect to 127ka, but we have also added this information to lines 112-113 for clarity, as recommended by the reviewer:

"In Section 3.1 we analyze the mean climate of the piControl. We then compare the climate of the LIG (127ka) to the PI (piControl), primarily focusing on Antarctica and the Southern Ocean. All anomalies are computed as the difference between 127ka and PI (difference = 127ka - piControl). Climatologies are computed using the final 50 years of the simulations. Section 3.2 examines the impact of the freshwater forcing on the climate response in the Southern Ocean, and the implications for AIS mass loss. All anomalies in Section 3.2 are computed between the

climatological 127ka mean and the evolving 127kaFW experiment (difference = 127kaFW - 127ka).”

Lines 130-132: can you show the increase in the westerlies in your 2degree simulation too? In Otto-Bliesner et al., 2020 this point is indeed made but I could not find a figure showing the stronger westerlies (I have not checked if there are SI figures, but there is no mention of this in their paper).

The reviewer is correct, there is no figure explicitly showing the increase in Westerlies in the N. Atlantic in the main text or supplement of Otto-Bliesner et al. (2020). However, we show this result below from the 127ka-1degree run in Otto-Bliesner et al. (2020).

[Figure]

Furthermore, we can confirm the effect is true in our 2-degree run as well.

[Figure]

Given that this is not a key result in our analysis, we have decided not to include a figure. This is in an effort to not grow the Appendix by too much, as we have added others during these revisions. Instead, we have changed the text here so that readers now distinguish that this result was discussed in Otto-Bliesner et al. (2020), but not explicitly shown.

Line 133: "(..more vigorous transport, not shown)" I suggest you include this in your Appendix, this is a rather crucial aspect for your following analysis of the SO (see also my comment about this at line 189).

Good point. We have now added the following figure to the Appendix and the text has been updated accordingly:

"Global and Atlantic northward heat transport is generally greater at all latitudes in the Northern Hemisphere in the *127ka* simulation compared to the *piControl* (i.e., more vigorous transport) (Fig A4). Southward heat transport in the Southern Hemisphere is weaker North of ~55°S, and slightly stronger South of ~55°S in *127ka* compared to *piControl* (Fig. A4). The North Atlantic annual upper cell (calculated as the maximum streamfunction found north of 28°N, and below 500m depth) strengthens in the *127ka* compared to *piControl* by about two Sverdrups, not shown). This is consistent with cooler North Atlantic sea surface temperatures (Fig. A3), as deep water formation increases and dense surface waters are cooled and brought to depth.  Otto-Bliesner et al. (2020) provides more detail on the barotropic streamfunction and AMOC changes during *127ka*."

[Figure]

Line 134: where is this shown?

Line 134 said: "The AMOC strengthens, as indicated in both the upper NH cell and deeper SH cell during 127ka compared to piControl (~2 Sv increase)."
See response to previous comment for the updated text that addresses this.

Lines 168-169: important point, if robust, add to conclusions(?)

The sentence referenced says: "Given this emerging new work, Antarctic temperatures in our 127ka simulation may not be inconsistent with proxy data (Figure 2a), though are probably on the low end." Because some of this key work is unpublished, we think it is premature to call this conclusion 'robust'.

Line 170: is the regional cooling in these sectors caused by sea ice changes?

The regional patterns are consistent between cooler SSTs and increased sea ice concentrations in the Ross, Amundsen and Weddell Seas and vice versa for the Pacific ocean sector. We discuss the connection between the weakened easterlies, the decreased sea ice and increased ocean temperatures a few paragraphs later:

"The greatest weakening of westerlies during the LIG is in the Atlantic–Indian Ocean sector (Fig. 4e). This region coincides with decreasing minimum sea ice extent and the greatest ocean warming both at the surface and throughout the ocean column, as seen in the MOT (Fig. 4c)."

We also have now added an explicit statement in (original) line 170 connecting this explicitly: "Southern Ocean SST's are warmer on average in 127ka compared to piControl, with some regional cooling such as in the Weddell, Amundsen, and Ross Seas (Fig. 2). Regional SST cooling (warming) coincides with sea ice concentration increases (decreases) (Fig. 4a)."

Line 189: SST warm anomalies can be linked to the LIG insolation forcing, but what is the dynamical link between LIG forcing and SO warming throughout the water column? Is it maybe the increased global oceanic heat transport under LIG forcing discussed in Otto-Bliesner et al. (2020) that was mentioned earlier? Is the increase in ocean heat transport for the LIG versus the PI shown in Otto-Bliesner et al. (2020)?

In the original manuscript, we showed weaker upwelling in the 127ka simulations near the AIS, and suggested this was a likely reason for the warmer subsurface temperatures (reduction of upwelling of cooler waters to this section of the water column (more on this in later responses to the reviewer)).   We have replaced this statement with a more complete explanation, as there are a number of interrelated mechanisms that lead to subsurface warming near the AIS under 127ka orbital forcing.

Recent work by Yeung et al. (2024) explored this very question, with similar simulations (127ka and piControl), using a different model (the Australian Community Climate and Earth System Simulator ACCESS-ESM1.5 (Ziehn, T. et al. 2020) at similar resolution. Yeung et al. find both the same sign of wind changes (reduced large-scale westerlies) and reduction in deep convection as in our experiments (see Fig below), and also note the possible role of changes in sea-ice production contributing to increased stratification and subsurface warming.  Yeung et al.  find greater warming than in our experiments (as much as 0.7°C in the Amundsen Sea), but following the same overall pattern.

[Figure]

**Figure**: Global stream function in PI climatology (left), 127ka climatology (center) and 127ka – PI (right).  Positive (negative) values indicate clockwise (counter-clockwise) circulation. The lower overturning cell located between 25 - 50S at about 3000-4000m depth weakens in the 127ka compared to the PI.

Furthermore, in our 2-degree 127 ka simulation, the westerlies shift equatorward, weakening the ACC. Meanwhile, easterlies closer to the continent increase (Fig 4e).  The same response is found in the ACCESS runs (Yeung et al 2024), which they argue induces a westward current off the WAIS coast, bringing relatively warm water to the Amundsen and Bellingshausen Seas..

We have now added the following language in our Result section, to summarize the above points:

"Seasonality of Antarctic sea ice is generally reduced in 127ka. The 127ka run simulates a reduction in Antarctic maximum sea ice extent, and a slight increase in minimum Antarctic ice area (Fig. A5). Annual mean Antarctic sea ice extent over the full 1000 year integration is reduced by about 0.4 million km$_2$ in 127ka compared to piControl."

"There are several interrelated mechanisms that lead to subsurface warming near the AIS during 127ka. Our simulation shows a reduction in the strength of the lower cell of the overturning circulation (see e.g. Marshall et al., 2012), and decreased ventilation.  The same response has also been shown recently by Yeung et al. (2024), who conducted a similar 127ka experiment using a different model (the Australian Community Climate and Earth System Simulator (Ziehn et al. (2020)) at similar resolution.  Yeung et al. (2024) find both the same sign of wind changes (reduced large-scale westerlies) and reduction in deep convection as in our experiments, and also note the possible role of changes in sea-ice production (also consistent with our CESM2 results) contributing to increased stratification and subsurface warming.  Yeung et al. (2024) find greater warming than in our experiments (as much as 0.7°C in the Amundsen Sea), but following the same overall pattern. We note that although our findings and those of Yeung et al. (2024) are consistent with one another, the relationship between wind forcing and ocean response is resolution-dependent, and future work with high-resolution models is likely to yield different results (Stewart and Thompson, 2012)."

References:

Marshall, John, and Kevin Speer. "Closure of the meridional overturning circulation through Southern Ocean upwelling." *Nature geoscience* 5, no. 3 (2012): 171-180.

Yeung, Nicholas King-Hei, Laurie Menviel, Katrin J. Meissner, Dipayan Choudhury, Tilo Ziehn, and Matthew A. Chamberlain. "Last Interglacial subsurface warming on the Antarctic shelf triggered by reduced deep-ocean convection." *Communications Earth & Environment* 5, no. 1 (2024): 212.

Ziehn, Tilo, Matthew A. Chamberlain, Rachel M. Law, Andrew Lenton, Roger W. Bodman, Martin Dix, Lauren Stevens, Ying-Ping Wang, and Jhan Srbinovsky. "The Australian earth system model: ACCESS-ESM1. 5." *Journal of Southern Hemisphere Earth Systems Science* 70, no. 1 (2020): 193-214.

General comment on section 3.1.2: up to this point in the paper changes between mean state PI and LIG are described but not discussed, why are we seeing these changes? what is driving them? why is the ocean warmer at depth during the LIG? why are the westerlies weaker?

Please see the response above for a response on this.

Figure 4, caption: "Climatologies are computed over the last 50 years of each simulation." You can omit this, as it is in your methods.
Omitted.

Figure 5, panel e: I do notice at the very beginning of your 127kaFW-SH timeseries a small bump, suggesting SH sea ice initially increased before starting declining. Have you looked into this? (I am asking out of personal curiosity). It is difficult for me to appreciate from this graph how many years it took for the sea ice to begin melting. This might be the transient sea ice response that we discussed in Guarino et al., 2023, and the subsequent decline in your simulation might confirm the two time-scales response of sea ice first proposed by Ferreira et al. 2015 and then invoked by Guarino et al. 2023 to explain the increase of Antarctic sea ice under H11 forcing.

We did indeed look at the sea ice response in the immediate few hundred years after the freshwater hosing began, and did not find any increase in annual sea ice area in the Antarctic (Fig below).  In fact, Antarctic sea ice area appears to remain fairly steady for about a century before beginning its decline. So, it appears that while this is a response in the HadGEM simulations, it is not so in the CESM2 simulations.

[Figure]

**Fig:** Initial 250 year response in sea ice extent for Antarctic (left) and Arctic (right). No apparent increase in SH sea ice extent is seen in the 127kaFW simulation. An immediate increase in Arctic sea ice extent is seen, a doubling in area in under a century that is maintained thereafter.

Line 225: should "127ka" be 127kaFW?
Yes - good catch.

Figure 7, panel a: "over the Antarctic continent" How is this defined? land points only?
Yes this is over land points. The caption has been modified to clarify this.

Figure 8: This figure is key to the paper results, and I would like to understand it better. In particular, I would like the authors to better substantiate some of their analysis. There will be some repetition between my remarks here and some comments below.

We will consolidate our answer to this, and the several follow-up questions that are listed immediately below as a group, since they all tie together.

If we look at the Weddell and the Ross/Amundsen Sea sectors, they behave in opposite ways, and I feel this aspect is not sufficiently discussed in the manuscript. Even more important, I don't understand why the authors talk about upwelling of cold waters (line 359). Generally, an increase in surface winds (and particularly in the negative wind curl in the SH) means, yes, increased upwelling (because the flow is divergent at the surface) but of relatively warm waters. This is a well-known mechanism for triggering open ocean deep convection and polynyas formation in CMIP models in the Weddell Sea. In fact, if I had to guess, I would think that the weaker (with time) winds and wind stress over the Weddell weaken the Weddell Gyre and discourage the occurrence of open ocean deep convection in the FW experiment compared to the LIG. Less Deep convection could imply less warm water entering the 200-800m water column, and thus the negative MOT anomaly persists in the Weddell.

The same is not true for the Ross Sea. In the Ross, as I also say in my comment at line 281-283, I can't really appreciate a strong trend in wind stress curl, but if there is one (as I think the authors say) then I would expect the same mechanism as above to be active: upwelling of warm waters as the Ross gyre spins up. Some models (not all) do simulate deep convection also in the Ross Sea (see de Lavergne et al., NCC, 2014) although I don't know where CESM stands on this.

Is it possible that the upwelling of warm waters in your bottom panels is linked to this? Finally, I believe the establishment of the initial MOT negative anomaly needs explaining.

We thank the reviewer for this line of questions. We hope to clarify these issues for all readers.

To start, the main issue to address is the mechanism of subsurface cooling in the freshwater simulation. In the original manuscript, we attribute the initial subsurface cold anomalies near the AIS to be a result of wind-driven upwelling. We show that peak westerlies shift poleward and wind stress curl strengthens. We recognize that the term upwelling may be confusing because this generally refers (in the Antarctic context) to the upwelling of CDW towards the surface. Here, we are referring to greater depths. What is important here is how changing wind stress influences the tilt of the isopycnal surfaces (Gregory, 2000; Marshall and Speer 2012)

To clarify how the isopycnals and temperatures shift in the first few hundred years of the 127kaFW run, we plot the background temperature state (colors) and isopycnal surfaces (grey scheme contours) (figs below). We do this for an example cross-section at 30E, but the story remains unchanged for the zonal mean average, suggesting that this is a robust response in all regions around the continent. The figure shows that within the first few hundred years, the isopycnals shift deeper in the column, and also steepen between 50S and 60S. At the same time, colder temperatures rise upward in the column, particularly near the continent. This is especially evident in the difference plot (right), showing the 127kaFW 300 years into the simulation minus the first 50 years of the 127kaFW run. Colder temperatures are upwelling along isopycnals from depth and brought toward the continent exactly in the depth range with which the ice sheet grounding lines are concerned.

[Figure]

Fig: Temperature (colors) and isopycnal surfaces (white-grey-black contours) at the beginning of the run (first 50 years mean) (left), a few centuries in (300-350year mean) (center) and difference between the two (right). In the difference plot, solid lines show isopycnal surfaces at the start of the run, and dashed lines show new position after ~300 years of simulation time.

In terms of the reviewer's comments regarding regional differences and trends beyond the first 1000 years, we do not attempt to disentangle these processes in this paper. While there is much to delve into here, this is beyond the scope of this current analysis.

Regarding the reviewer's comment on the relationship between deep ocean convection and subsurface water temperature, please see our response above.

We have incorporated this more thorough explanation of mechanisms into our revised manuscript. We have also included the new figure shown above.

In the Results, we now discuss the new Figure:

"Furthermore, over the course of the first few hundred years of the 127kaFW simulation, the isopycnals shift deeper and steepen between 50 and 60S (Fig. 10). As a result, colder temperatures are drawn upward along isopycnals from depth and brought toward the continent at the depth ranges most relevant to the ice sheet. Fig. 10 shows an example cross-section at 30E; the same picture applies to other locations and in the zonal mean."

We have added some text to the Conclusions to also note the findings of future simulations that produce a similar subsurface cooling response in the high latitude southern ocean:

"Interestingly, simulations of future changes in the Southern Ocean show that increasing SH westerlies lead to subsurface cooling near the AIS, driven by similar mechanisms that we detail in our 127kaFW experiment. In an idealized wind-forcing experiment, Armour et al. (2016) use an ocean model (MITgcm) and finds SST increases and subsurface cooling on the order of a few tenths of a degree in the Southern Ocean (their Fig S12) in response to increased westerly winds. In an analysis of 19 climate models under a future warming scenario, Yin et al. (2011) show subsurface cooling in the high latitude Southern Ocean. They attribute these changes to a southward shift of the westerlies, intensification of the ACC, Ekman-induced upwelling of cold deep waters and blocked propagation of ocean warming signals."

We have also changed the language in other locations in the conclusions to reflect the new analysis:

"The initial subsurface ocean cooling in our 127kaFW simulation is related to wind-driven changes in ocean circulation as a result of freshwater flux in the North Atlantic. As peak westerlies shift southward, isopycnals shift deeper and become steeper, drawing colder water from depth along isopycnals up to depths relevant to the ice sheet."

**References:**

Gregory, Jonathan M. "Vertical heat transports in the ocean and their effect on time-dependent climate change." *Climate Dynamics* 16 (2000): 501-515.

Marshall, John, and Kevin Speer. "Closure of the meridional overturning circulation through Southern Ocean upwelling." *Nature geoscience* 5, no. 3 (2012): 171-180.

Yin, Jianjun, Jonathan T. Overpeck, Stephen M. Griffies, Aixue Hu, Joellen L. Russell, and Ronald J. Stouffer. "Different magnitudes of projected subsurface ocean warming around Greenland and Antarctica." *Nature Geoscience* 4, no. 8 (2011): 524-528.

Crowley, T. J. and Parkinson, C. L.: Late Pleistocene variations in Antarctic sea ice II: effect of interhemispheric deep-ocean heat exchange, Clim. Dynam., 3, 93–103, 1988.

Renssen, H., Goosse, H., Crosta, X., and Roche, D. M.: Early Holocene Laurentide Ice Sheet deglaciation causes cool- ing in the high-latitude Southern Hemisphere through oceanic teleconnection, Paleoceanography, 25, PA3204, https://doi.org/10.1029/2009PA001854, 2010.

Line 264: "Within the first 500 years, a bulwark of cold subsurface water wrapping around the continent" What is driving this?

We think our language here was misleading, and we want to clarify. We recognize that it implied that there was a cold water influx, rather than just an anomaly with respect to 127FW. It has now been revised to the following: "Within the first 500 years, MOT around the continent decreases with respect to the control case (127ka)."   The response to the previous comment addresses the mechanism behind this more fully.

Lines 280-281: is this shown? does this mean that the SAM remains more positive than the LIG during the whole LIG-FW experiment?

We show anomalies in wind vectors in Figure 8, second row (quivers), which is now clarified in the text. However, we do not actually show the southward shift of the mean zonal winds by several degrees.  The text is now clarified to make this point clear:

"The peak Southern Hemisphere westerlies strengthen and shift south by several degrees in the first few hundred years following freshwater forcing (not shown), and remain in their poleward-shifted position for the rest of the simulation (quivers, Fig. 8, second row)."

For the interest of the reviewer, below we have plotted zonal mean winds for the freshwater forcing experiment (labeled H11) at different times of the experiment, as compared to the climatological mean winds for the 127ka control run. This also indicates the southward shift of the westerlies that is sustained at a fairly constant latitude for the rest of the simulation.

[Figure]

Lines 281-283: Okay the physical mechanism, but in Fig.8 the wind stress weakens with time, particularly in the Weddell. As for other regions, Ross and AS, if there is a trend (i.e. wind curl becoming more and more negative) this cannot be appreciated from Figure 8 where I see almost no difference from 500 to 4000yrs.

We appreciate the reviewer's attention to detail here. While we find the multi-millennial trends on regional scales interesting, there are likely a host of locally-controlled coupled interactions at play here. Therefore, we prefer to keep the analysis in the paper limited to the zonal mean.

We have therefore updated the text here to read:

"As the westerlies move southward, a band of negative wind stress curl anomalies encircles the continent (blue shades, Fig. A7, second row). This shift of westerlies ramps up over the first 500 years of the simulation, and then remains fairly steady in magnitude and position thereafter (Fig. A7)."

For the curiosity of the reviewer, the plot below shows the wind anomalies between 127kaFW - 127kaControl, 50 year average snapshots, every 250 years:

[Figure]

In Fig A7, which we now reference here, we are showing how the westerlies shift south, ramping up over the first 500 years. The trend in u-winds during this period is perhaps weakest in the Weddell, but the trend is nonetheless in the same direction.

We also do not see an appreciable trend beyond the first 500 years of the simulation. The plot below shows the *trend* in u-wind anomalies (127kaFW - 127ka) over the course of the first 500 years of the simulation (top row), as well as for the rest of the simulation (500 - 4000 years) (bottom row). From this, we see that the Weddell Sea shows weaker trends in increasing westerlies in the first 500 years than other regions around the AIS, but the trends are still positive. After 500 years, there is no appreciable trend in westerlies either increasing or decreasing in this region, suggesting that the new location of the winds remains more or less consistent.

[Figure]

Furthermore, the reviewer's reference to Fig 8 shows the wind stress snapshots at 1000 year intervals. While it may appear in these snapshots that the Weddell wind stress curl is weakening, we find that this is more likely a result of sampling the variability in the system, and we caution against over-interpretation.

Lines 286-287: I feel that the reasons for the subsurface cooling in the SH are not well explained anywhere in the manuscript, if this has been shown in other studies references plus a short description of the physical mechanisms at play must be included.

Please see our full response to this issue above.

Line 291: has 'CDW' been defined before?

It has now been defined here. Thank you.

Line 338: Is this truly unexpected? There is some literature about this.

Guarino et al., 2023 (as you mention later on) and some of the references in there provided:

Crowley, T. J. and Parkinson, C. L.: Late Pleistocene variations in Antarctic sea ice II: effect of interhemispheric deep-ocean heat exchange, Clim. Dynam., 3, 93–103, 1988.

Renssen, H., Goosse, H., Crosta, X., and Roche, D. M.: Early Holocene Laurentide Ice Sheet deglaciation causes cooling in the high-latitude Southern Hemisphere through oceanic teleconnection, Paleoceanography, 25, PA3204, https://doi.org/10.1029/2009PA001854, 2010.

As far as we can tell, this modeled response of anomalous cooling subsurface temperatures in the high latitude southern ocean after a freshwater hosing experiment is robust. However, the literature fails to highlight or discuss the mechanisms behind this response. Guarino *et al.* (2023) certainly sees this response in Figure 3b, and as we discuss in our manuscript, it is seen in other runs such as the iTrace runs, Yin et al (2011) and Armour et al (2016).  In any case, we have removed the word 'unexpected' here.

Lines 341-342: I agree on this.

Great.

Lines 353-355: Well, this point exactly is the one we make in Guarino et al. 2023, in which the short timescale response is analysed and where HadCM3 results are invoked to explain the long time scale response.

It appears that our sea ice response is more consistent with the older HadCM3 sea ice response in which sea ice does not show a short-term expansion in the southern hemisphere. We've discussed this a little more in a previous response.

In fact, it would be quite interesting to take a look at your first 200-300 years of simulations and see if results are consistent with HadGEM3 ones, confirming the transient sea ice response.

We have looked into this, as discussed above. We do not see the same transient sea ice response that Guarino et al. 2023 saw in HadGEM3.

Line 370: Again, I feel this dynamical response of the system ywell explained. Why does this happen in the LIG experiment?

As noted in the above responses to the reviewer, we have now added in more discussion of mechanisms in the manuscript now. Please see responses above for more on this.

Lines 385-389: This paragraph seems a repetition, the Discussion section begins in the same way. Consider keeping only one of the two.

Thanks, this has been removed from the conclusions now.

Maria Vittoria Guarino

**Reviewer 2**

Review of "Antarctic climate response in Last-Interglacial simulations using the Community Earth System Model (CESM2)"
by Mira Berdahl, Gunter R. Leguy, William H. Lipscomb, Bette L. Otto-Bliesner, Esther C. Brady, Robert A. Tomas, Nathan M. Urban, Ian Miller, Harriet Morgan, and Eric J. Steig

The climate of the Last Interglacial was characterized by warmer temperature and higher sea level, in response to different orbital parameters. The sea level rise was likely partly due to a partial West Antarctic ice sheet melt, possibly due to Southern Ocean warming. The ocean warming could be related to an AMOC weakening induced by fresh water originating from Northern ice sheet melting. To better understand what could have caused an Antarctic ice sheet melt, Berdahl et al. analyse two CESM2 simulations run with 127ka orbital parameters, including one with fresh water fluxes. The simulations are compared to data with a focus on Antarctica to evaluate the possibility of favouring an Antarctic ice sheet melt.

Neither of the two simulations capture the full magnitude of the temperature increase at 127ka recorded in proxy data, but the 127ka simulation shows a small ocean temperature increase that could favour West Antarctic melt. The 127ka simulation with fresh water fluxes displays a local cooling around Antarctica despite general warming in the Southern Hemisphere, indicating that this is unlikely to explain the Antarctic ice sheet melt according to CESM2.

The paper is well written and well organized, and the two simulations are interesting. But this study is relatively limited. It would have been more comprehensive and useful to test the impacts of Antarctic changes such as local freshwater fluxes and a smaller ice sheet to evaluate the impact on climate and possible feedbacks.

We thank the reviewer for their comments on this paper. We agree with the reviewer that simulations that test the effects of freshwater feedbacks and different ice sheet configurations would be relevant to this study. However, the focus of our paper is on long (multi-millennial effects) of solar and North Atlantic freshwater forcing on southern ocean conditions near the AIS. Since these are coupled global simulations, each simulation costs over 2 million core

hours and takes several months to run.  Therefore, it is beyond the scope of this project to include and run further simulations.  Below we have responded more specifically to all comments.

**General comments**

Since the focus is the Antarctic climate and the possibility of the Antarctic ice sheet collapse it would have been interesting to test the impact of fresh water flux from Antarctic, whether this could trigger a positive feedback. There is a small discussion on it, this could be developed.

We agree this is an interesting area of study that prompts more investigation by the community.  As the AIS continues to melt and release freshwater along its periphery, ocean density and circulation will inevitably be impacted.  Though it would be very useful for the 127ka-H11 experiment, it is beyond the scope of this study to run any further coupled simulations. However, we have developed this discussion a little further by discussing results from CESM1 simulations that directly test inputs of freshwater both at the surface and deeper in the column.  We have now added the following text:

"Studies using CESM (Version 1) specifically test the impacts of AIS freshwater discharge by inputting freshwater at the ocean surface as icebergs (Pauling et al., 2017) or at the depth of the ice shelf fronts around the continent (Pauling et al., 2016). In both cases, the model generates subsurface warming near the AIS due to increased stratification of the water column and subsequent reduction of sinking of cold continental shelf waters."

The papers referenced are:
https://doi.org/10.1175/JCLI-D-15-0501.1
https://doi.org/10.1002/2017GL075017

On the same topic, it would have been great to test the impact on climate of having a smaller Antarctic ice sheet and the possible feedback through ocean temperature and surface mass balance changes.

We agree with the reviewer that this is an interesting set of problems to examine. Others (e.g., Tewari et al. (2021a), Tewari et al. (2021b), Steig et al. (2015)) have explored the effect of different (reduced) ice sheet configurations on atmospheric circulation.  However, in order to properly capture feedbacks between ice sheet size, ocean, and atmosphere, these components must be fully coupled in the model. Unfortunately, this capability is not yet possible with CESM2, and there are perhaps only one or two other models in the world that have a fully coupled Antarctic ice sheet component in their earth system model.

Finally, the fresh water flux addition results in large changes in disagreement with proxy data, but a smaller fresh water flux could improve the model-data comparison.

We could speculate on this in the manuscript, but part of our argument here is that perhaps we do not need the freshwater forcing at all in order to explain the proxy records.

Would it be doable to do one (or several) additional simulations? How long does it take to run 1000 years? Additional simulations could be shorter than the ones presented here, even if the equilibrium is not reached it would still be interesting.

Generally, at this resolution, it takes from one to three months of real time to run 1000 years of the coupled simulation. Furthermore, it costs about 2 million core hours to do this. This is well beyond the scope of what we have available, and while we agree it would be interesting, this would have to be reserved for future work.

Three complementary simulations might be worth considering:
- With 127ka ice sheets / smaller West Antarctic ice sheet (this could have a regional impact)
- With fresh water fluxes in the Southern Ocean
- With less intense fresh water fluxes in the North Hemisphere
and possibly a combination of the three.

Unfortunately, as we explain above, these are all beyond the scope of this work as more simulations are not computationally possible at this time and would be a great topic for a follow-up study. Other work is directly targeting the questions of impact of freshwater flux from AIS to the ocean – (SOFIA intercomparison for example).

Since the number of figures is reasonable, and looking at the additional figures while reading the manuscript is not very comfortable, I would advise to switch some of the additional figures to the main text, especially those including Southern Ocean temperature (as this is the most relevant for possible Antarctic ice sheet melt) such as figures A2, A3 and A5.

In response to other reviewers, we have now added several more figures to the main text and to the supplement. As a result, we have decided to keep the original supplement figures as-is, in order to not overwhelm the main manuscript.

**Specific comments**

Abstract

Could you include a sentence on the comparison with data and which simulation is best, or more likely?

We emphasize that our purpose here is to evaluate mechanisms, rather than focus on a specific model/data comparison. Given the uncertainties in the data, and the rather idealized nature of our experiments, we prefer not to make this statement in the abstract.

Model description

Line 67 explain what is FV1? 1-degree model? It is mentioned as the 1-degree model but only much later (p.19). How long are the simulations? What is the gain compared to the 1degree resolution version?

We have added clarification here on what FV1 means (with approximately 1deg resolution in atmosphere and ocean). The simulation lengths are all given in Table 1. The gain is largely in computational cost. For example, at 2-deg resolution we can run our freshwater experiment for 4000 years, which would be difficult (in human time and storage) with a 1deg resolution.

Results

Line123: "the orbital-only forcing underestimates" -> the orbital-only simulation underestimates

Thanks, done.

Line 125-126: can you give the temperature change in CESM 1degree to compare with? And also, you could compare to other models as described in Otto-Bliesner 2021.

We have added some text to put this into context with the Otto-Bliesner 2020 results with the FV1 simulation:

"Otto-Bliesner (2020) find that in their CESM2-FV1 simulation, global annual mean temperatures are actually cooler by 0.1C in 127ka compared to piControl. They find that while JJA near-surface temperatures are 1.09C warmer in the 127ka compared to piControl, DJF temperatures are colder by nearly 1C."

Comparing to other models is beyond the scope here, but we refer the reader to the Otto-Bliesner (2020) paper for more information.

Line134-135: are you showing this (AMOC change) somewhere? Or are you referring to another paper? Is the AMOC change the same in the two model versions?

This was referring to results from our simulations, but the reviewer is correct that we didn't show this in our figures. We have changed the text to clarify this:

"Global and Atlantic northward heat transport is generally greater at all latitudes in the Northern Hemisphere in the *127ka* simulation compared to the *piControl* (i.e., more vigorous transport) (Fig A4). Southward heat transport in the Southern Hemisphere is weaker North of ~55◦S, and slightly stronger South of ~55◦S in *127ka* compared to *piControl* (Fig. A4). The North Atlantic annual upper cell (calculated as the maximum streamfunction found north of 28◦N, and below 500m depth) strengthens in the *127ka* compared to *piControl* by about two Sverdrups, not shown). This is consistent with cooler North Atlantic sea surface temperatures (Fig. A3), as deep water formation increases and dense surface waters are cooled and brought to depth. Otto-Bliesner et al. (2020) provides more detail on the barotropic streamfunction and AMOC changes during *127ka*."

Figure 3. What is the variable plotted in the background? Either specify the variable adding units and colorbar if it is useful to keep, or remove the background colors to only keep the different regions.

Thanks for the feedback. The background has been removed and the figure updated.

Section 3.2.2
The eddy component seems to play an important role. Can you elaborate on how eddies are represented and how this could change (or not) in a higher resolution model?

The eddies are represented by the Gent-McWilliams parameterization (Gent & McWilliams, 1990). Performing simulations using a higher resolution ocean model at 0.1 deg would help us answer the question on the impact of the parameterization on our simulation. As such

experiments in similar conditions have not been run (to the best of our knowledge) we will not speculate on their outcome.

In other model experiments with freshwater fluxes (apart from CESM, and in other than LIG period), has this cooling been observed before? Is this robust or model dependent?

In our original manuscript we included discussion of other freshwater hosing experiments (e.g. iTrace runs, and other idealized experiments, for example Pedro et al. 2018) that exhibited subsurface cooling near the AIS. However, these papers did not get into the details of why this occurs or what the mechanisms and implications may be.  We have now expanded our discussion to include mention of subsurface cooling in other papers that show this response but are not simulating the LIG period.  The discussion also includes a more thorough dive into the mechanisms that drive these changes.  For more on the latter, please see responses to Reviewer 1.

We have added some text to the Conclusions to also note the findings of future simulations that produce a similar subsurface cooling response in the high latitude southern ocean:

"Interestingly, simulations of future changes in the Southern Ocean show that increasing SH westerlies lead to subsurface cooling near the AIS, driven by similar mechanisms that we detail in our 127kaFW experiment. In an idealized wind-forcing experiment, Armour et al. (2016) use an ocean model (MITgcm) and finds SST increases and subsurface cooling on the order of a few tenths of a degree in the Southern Ocean (their Fig S12) in response to increased westerly winds. In an analysis of 19 climate models under a future warming scenario, Yin et al. (2011) show subsurface cooling in the high latitude Southern Ocean. They attribute these changes to a southward shift of the westerlies, intensification of the ACC, Ekman-induced upwelling of cold deep waters and blocked propagation of ocean warming signals."

Discussion

Line 307-312. The bipolar seesaw response is too large with the imposed fresh water flux. Could you discuss the possibility of having a better response with a smaller fresh water? It would be interesting to have a simulation with a smaller fresh water flux to evaluate the climate Response.

The PMIP protocol for the 127kaFW simulations was to include 0.2Sv of freshwater flux.  The reviewer is correct that this is quite a bit of freshwater to add to the system, and in the future it may be advisable to consider smaller amounts of freshwater.  That said, the iTrace run does include smaller fluxes, but still shows a subsurface cooling in response.  We discuss this in our manuscript.

Line 329- 333. It would have been great to have a simulation with a fresh water flux in the Southern Ocean to evaluate its impact.

We agree, and expect that these types of experiments will be part of future work with CESM2 (either through participation in the SOFIA intercomparison project, or other endeavors such as WhatIfMIP).  We have elaborated on this topic in our Discussion.

Line 337- 342. You could discuss the implication in terms of transient evolution if this cooling and delayed warming is a robust feature: could it result in preventing the Antarctic ice sheet

from melting too early when the Northern ice sheets are melting, and allow the Antarctic ice sheet to melt later?

Given the rather idealized nature of our experiments, we would prefer not to speculate on these implications. There is surely more to do in another, future paper, but it would require additional experiments.

Conclusions

Do you have ideas on what could be missing to obtain temperature changes in better agreement with proxy data?

This is difficult to answer as in some areas we do match proxy data well and some, less so. Proxy data have their own source of errors which can also lead to mismatches. We would caution against trying to match the proxy data.. It would be interesting to pursue future work using methods such as data assimilation, but this is beyond the scope of our current work.

**References**

Tewari, K., Mishra, S.K., Dewan, A. et al. Effects of the Antarctic elevation on the atmospheric circulation. Theor Appl Climatol 143, 1487–1499 (2021). https://doi-org.cuucar.idm.oclc.org/10.1007/s00704-020-03456-1

Tewari, K., Mishra, S. K., Dewan, A., Dogra, G., & Ozawa, H. (2021). Influence of the height of Antarctic ice sheet on its climate. *Polar Science*, *28*, 100642. https://doi.org/10.1016/j.polar.2021.100642

Gent, P.R., McWilliams, J.C., 1990. Isopycnal mixing in ocean circulation models. J. Phys. Oceanogr. 20, 150–155

Steig, Eric J., Kathleen Huybers, Hansi A. Singh, Nathan J. Steiger, Qinghua Ding, Dargan MW Frierson, Trevor Popp, and James WC White. "Influence of West Antarctic ice sheet collapse on Antarctic surface climate." *Geophysical Research Letters* 42, no. 12 (2015): 4862-4868.

**Reviewer 3**

Berdahl et al. examine CESM2 model runs under pre-industrial and 127ka orbital parameters. The focus is mostly on the degree and mechanisms of ocean warming adjacent to the Antarctic ice sheet (AIS) in 127ka simulations with and without fresh water (FW) forcing. In particular, the paper proposes the hypotheses "that warming in the Southern Ocean, owing simply to the insolation anomalies during the LIG, may have been sufficient to cause substantial WAIS collapse". This is certainly a worthwhile topic to explore and the paper raises some new and interesting insights.

In my view there are two major results.

- There has been a common view that North Atlantic FW forcing of the magnitude of a H event (ca. 0.2 Sv) drives warming against the Antarctic ice sheet. However, Berdahl et al. show modest ocean cooling against the AIS in their 127ka FW simulation and give a sound treatment of what is driving the cooling. They also identify similar, largely overlooked, subsurface cooling in previous simulations where FW is applied to the NA.

-

- They identify modest sub-surface warming (0.4C), in the 127ka run without FW forcing and suggest that the warming is orbitally forced. Here, I was less clear on the specific processes driving the subsurface warming in the model and their link to orbital forcing.

In my view the paper makes some good and new advances on understanding of impacts of orbital and FW forcing on high southern latitudes. Congratulations on this large, detailed and well written study. I recommend publication after addressing the major and technical points below.

We thank the reviewer for their very kind and valuable comments and suggestions. We have addressed all of their points below, and believe these changes have improved the paper.

Major points:

Some more attention is needed to clarifying significance of anomalies in the text and in figures, refer to technical points below for specific examples.  This has been done. Please see technical points below for morer details.

Concerning analogs.. Line 366: "Given that we do not expect a large freshwater forcing of comparable scale to the H11 event in the future, our 127ka simulation, without freshwater forcing, may be a more relevant analog for the future". I don't think the analog point is valid as written here and elsewhere in the text (e.g. line 411). The future has different orbital parameters and CO2 levels to 127ka. Also, there is growing evidence that an AMOC collapse this century is plausible, due to NA warming and freshening (e.g. Ditlevsen & Ditlevsen, 2023) and previous modelling indicates that the far-field impacts of AMOC collapse are similar irrespective if the trigger is FW or some other mechanism (e.g. Brown and Galbraith, 2016). Also relevant to the analog argument is the different (and changing) thermal structure of the ocean now compared to 127ka. The impact of changes in SO windstress and upwelling on subsurface temperatures near the AIS is sensitive to the temperature and properties of the upwelled waters, e.g. in the present climate where CDW is warming (e.g. Auger et al., 2021), there is evidence the poleward shifted westerlies are driving sub-surface warming, not cooling (e.g. Herraiz-Borreguero & Naviera Garabato, 2022). These factors need to be considered in commenting on the results and relevance of the current simulations for future AIS melt.

This is fair. The comments on analogs here have been removed. (Line 411 is addressed separately in a later response to the reviewer.) Instead we have emphasized that we only mean that the subsurface warming generated near the AIS in the 127ka is more in line with our expectations for the future, rather than the changes seen in 127kaFW.  We have also included language that AMOC collapse has been predicted this century and that the far-field

impacts are similar irrespective of the trigger (see response to comment about line 40, below).

Line 127 to 139: the text here begins to discuss but does not close out why global ocean heat content is higher in the 127ka simulation than in the PI. Similarly, Section 3.1.2 is compelling that there is subsurface warming close to the AIS in 127ka compared to PI, but I miss a clear explanation of what is driving the warming. Is it reduction in upwelling in the Tight SO? I think this is a major point because it goes to a key conclusion of the paper that its orbital forcing driving the sub-surface warming – some more explanation is needed to clarify the connection from orbital forcing to sub-surface warming near the AIS.

We agree that more explanation is useful for the 127ka subsurface warming.

In the original manuscript, we showed weaker upwelling in the 127ka simulations near the AIS, and suggested this was a likely reason for the warmer subsurface temperatures (reduction of upwelling of cooler waters to this section of the water column (more on this in later responses to the reviewer)).  However, we suspect there are a number of mechanisms that lead to subsurface warming near the AIS under 127ka orbital forcing.

Recent work by Yeung et al (2024) explored this very question, with similar simulations (127ka and piControl), using a different model (the Australian Community Climate and Earth System Simulator ACCESS-ESM1.5 (Ziehn, T. et al. 2020) at similar resolution. Yeung et al. find both the same sign of wind changes (reduced large-scale westerlies) and reduction in deep convection as in our experiments (see Fig below), and also note the possible role of changes in sea-ice production contributing to increased stratification and subsurface warming.  Yeung et al. also find greater warming than in our experiments (as much as 0.7°C in the Amundsen Sea), but following the same overall pattern.

[Figure]

**Figure**: Global stream function in PI climatology (left), 127ka climatology (center) and 127ka – PI (right).  Positive (negative) values indicate clockwise (counter-clockwise) circulation. The lower overturning cell located between 25 - 50S at about 3000-4000m depth weakens in the 127ka compared to the PI.

Furthermore, in our 2-degree 127 ka simulation, the westerlies shift equatorward, weakening the ACC. Meanwhile easterlies closer to the continent increase (Fig 4e in the manuscript).  The same response is found in the ACCESS runs (Yeung et al 2024), which they argue induces a westward current off the WAIS coast, bringing relatively warm water to the Amundsen and Bellingshausen Seas. We suspect a similar mechanism is at play in CESM2.

We have now added the following language to summarize the above to the Results:

"Seasonality of Antarctic sea ice is generally reduced in 127ka. The 127ka run simulates a reduction in Antarctic maximum sea ice extent, and a slight increase in minimum Antarctic ice area (Fig. A5). Annual mean Antarctic sea ice extent over the full 1000 year integration is reduced by about 0.4 Million km$_2$ in 127ka compared to piControl."

"There are several interrelated mechanisms that lead to subsurface warming near the AIS during 127ka. Our simulation shows a reduction in the strength of the lower cell of the overturning circulation (see e.g. Marshall and Speer (2012)), and decreased ventilation.  The same response has also been shown recently by Yeung et al. (2024), who conducted a similar 127ka experiment using a different model (the Australian Community Climate and Earth System Simulator (Ziehn et al., 2020)) at similar resolution.  Yeung et al. (2024) find both the same sign of wind changes (reduced large-scale westerlies) and reduction in deep convection as in our experiments, and also note the possible role of changes in sea-ice production (also consistent with our CESM2 results) contributing to increased stratification and subsurface warming.  Yeung et al. (2024) find greater warming than in our experiments (as much as 0.7°C in the Amundsen Sea), but following the same overall pattern. We note that although our findings and those of Yeung et al. (2024) are consistent with one another, the relationship between wind forcing and ocean response is resolution-dependent and future work with high-resolution models is likely to yield different results (Stewart and Thompson, 2012)."

References:

Marshall, John, and Kevin Speer. "Closure of the meridional overturning circulation through Southern Ocean upwelling." *Nature geoscience* 5, no. 3 (2012): 171-180.

Yeung, Nicholas King-Hei, Laurie Menviel, Katrin J. Meissner, Dipayan Choudhury, Tilo Ziehn, and Matthew A. Chamberlain. "Last Interglacial subsurface warming on the Antarctic shelf triggered by reduced deep-ocean convection." *Communications Earth & Environment* 5, no. 1 (2024): 212.

Ziehn, Tilo, Matthew A. Chamberlain, Rachel M. Law, Andrew Lenton, Roger W. Bodman, Martin Dix, Lauren Stevens, Ying-Ping Wang, and Jhan Srbinovsky. "The Australian earth system model: ACCESS-ESM1. 5." *Journal of Southern Hemisphere Earth Systems Science* 70, no. 1 (2020): 193-214.

Technical Points:

Global mean SAT in the 127ka is described as 0.004°C warmer than PI. Please clarify if this difference is significant using an appropriate test. I suspect it may not be, either way it should be made clear in the text. If the answer is that it's not significant, then 'marginally warmer' should not be reported in the abstract. Instead report the significant seasonal temperature anomalies.

The 0.004C anomaly is statistically significant to the 95% confidence interval, if computed over the entire 1000 year period. If it is computed over only the final 50 years, as the climatological mean is computed, then it is found to be insignificant.  This is now noted in the Table 2 caption.

Line 29: Maybe it's a common view, but I don't think it is canonical.

Agreed - this language has been modified.

Line 36: Give the estimated timing of the FW discharge. Also give the timing of the LIG sea level maximum (ca. 125ka?).

Done.

Text has been modified now to read:

"Sea-level records from the LIG indicate that global mean sea level was 4--9 m higher than present, peaking sometime after ~125 ka (Dutton et al., 2015)."

And

"A large freshwater discharge, associated with the the Heinrich-11 (H11) event, is known to have occurred a few thousand years prior to the LIG (Bohm et al., 2015) (estimates suggesting it occurred between 135 ka and 130 ka (Clark et al., 2020)), consistent with this idea."

Line 40: But you should also note that AMOC collapse has been predicted this century, and the far field impacts are similar irrespective of trigger. See e.g. Ditlevsen & Ditlevsen (2023) and Brown and Galbraith (2016).

This is true. We have changed the text now to include this.

Even though a freshwater discharge event comparable to that of H11 is highly unlikely to occur in the present-day climate, AMOC collapse has been predicted this century due to North Atlantic warming and freshening (Ditlevsen and Ditlevsen, 2023), and far-field impacts may be similar irrespective of trigger (Brown and Galbraith, 2016).

Line 55: Give the volume in m. sea level equivalent for context.

The conversion to mm/year rate of SLR has been added.

The model set up and experimental design are appropriate and well described.

Thank you!

Table 2, denote which anomalies are significant at 95%CI.

We have now noted in the Table 2 caption that all anomalies aer found to be significant at the 95% confidence interval.

Line 99: 3 to 8 thousand years before..

Thanks, changed.

Excellent figures throughout!

Thank you!

Line 175 to 180 and elsewhere, clarity about significance of anomalies is needed. Probably all these anomalies are significant given the 1000 year run times. But this needs to be tested and stated. Similarly, Fig 4. marks significance with hatching for the snowfall rate but not for the other panels.

We have confirmed and stated that the significance of the anomalies in Table 2, Table 3, Fig 2 are statistically significant to the 95% confidence level.

In Figure 4, all panels have been updated to include significance hatching.

Figure 6 shows shading with significance at 95%.

Captions for these Tables and Figures now include this information.

Line 178. Figure 2 misses a. b. etc labels.

Updated.

Line 189, and similar to Major Point 3):. "As noted, the Southern Ocean generally warms as a result of the orbitally-forced 127ka". If it's to be accepted that the warming is *the result* of orbital forcing then some more explanation and justification is needed closing the link between orbital forcing and the warming. If you can't get to the bottom of what processes are driving the warming that's ok, but it needs to be tackled head on (by comparison the explanation of how FW forcing drives cooling against the AIS is more thorough and complete).

This is a great point.  Another reviewer has also pointed this out.  We have addressed this in Major Point 3 above.

Fig 3. There is no colorbar here for the shading, presumably SST.

Thanks for the feedback. The background field has been removed so there is no need for a colorbar any longer. The figure has been updated.

Line 237: Quantify the cooling here.

Done

Line 248: all ocean basins accumulated heat..

Yes true, we've modified the text to clarify this.

Line 249: a ca. 500-yr cooling..

Changed, thanks.

Mechanistically the reason for different responses in the SO and 'Tight SO' is that the Tight SO is south of the ACC, which is a barrier to meridional heat transport. This point is relevant to the definition and should be made in the text.

This is now added to the definition of the TightSO in Figure 3.

Fig 5f and line 229: From the figure this looks more like no change in OHC than 'a sustained reduction'. Can you demonstrate more clearly that this is a sustained and a significant reduction? The point is better made by Figure 6 and 8 than 5f. Also, from Fig 7 we see an initial cooling for ca. 500- 1000 years followed by gradual warming. Hence some more nuance is needed than 'sustained reduction' and 'sustained cooling'.

Yes, we agree. The language has changed to no longer refer to a 'sustained cooling', and the reference to the figures has been updated to point to the Figures that better illustrate the evolution of the cooling.

Line 237. Rather than 'distinct cooling' give the delta T.

Done.

Fig 6. Are all shaded anomalies significant at 95%?

Yes, they are all significant to 95%.  We have added a note in the caption to clarify this.

Line 252-255etc: 'After this multi-century cooling, MOTs rebound slowly for the remainder of the simulation.' This is a clear description and should come earlier. Some of the earlier descriptions could even be trimmed.

Agreed, and the text has been changed in multiple places to improve this.

Line 261: Good point.

Thanks

Line 264: I don't think it's appropriate to describe this as a 'bulwark of cold subsurface water'; this is an anomaly with respect to 127Fw and not showing absolute water temperature and readers could miss the important distinction. Delete or revise.

Yes we agree - it created confusion for another reviewer as well. We have modified the sentence to clarify this: "Within the first 500 years, MOT around the continent decreases with respect to the control case (127ka)."

Lines 285 to 290 etc: Great analysis.

Thank you!

Line 297: Good to hear. But the phrasing 'is at least partly responsible for the slow rebound in the initial wind-driven cooling at depth near the AIS' may be confusing. Do you mean that the eddy driven transport is at least partly responsible for the longer-term warming trend that replaces the initial wind-driven cooling? (I think it is explained more clearly later at line 360).

Yes this is what we mean. We have updated the text to clarify here, thanks.

Line 322: The argument here is that 127ka orbital forcing results in warmer subsurface temperatures against the AIS than we see in PI and therefore orbital forcing plausibly drove LIG mass loss from the AIS. An alternative way to test this idea would be to examine how subsurface temperatures evolve using a time slice before the LIG with respect to the 127ka run. This could diagnose if there is an orbitally driven sub-surface warming *trend* across the LIG. I don't expect you to run new simulations, but please address this point in the text, and if your approach is the better way of diagnosing forcing by sub-surface warming then explain why.

Great point! We have added the following line to this paragraph to mention this would be a viable alternative method to evaluate subsurface warming from orbital changes.

"Future work might consider an analogous comparison between the 127ka and a preceding period (rather than the pre-industrial) in order to further evaluate and understand orbitally-driven subsurface warming."

Line 375-380: Add a citation to Herraiz-Borreguero & Naviera Garabato, Nature Clim. (2022), which explores these factors in modern observational data, including the relationship between sub-surface warming and the strength and position of the westerlies. Their observations link poleward shifted (DJF) westerlies to *warming* of mid-depth CDW at depths and locations consistent with AIS ice melt. This appears opposite to the trend in subsurface temperatures in response to stronger westerlies in 127kaFW simulation. I expect this is because in the present day the increased wind stress is drawing up anomalously warm CDW. I think it's worth to discuss this, given the paper's intent that results are relevant for understanding future forcing on the AIS.

Agreed – this is relevant literature to include here. We have added text to describe these efforts and how they fit into the current efforts to tease apart recent/future changes.

"Recent studies of the Amundsen Sea (e.g. Naughten et al., 2021) and East Antarctica (e.g. Herraiz-Borreguero & Naviera Garabato, 2022), for example, use high resolution models and observations to disentangle the complex relationship between wind forcing and sub-surface warming."

Line 407: A key result that deserves a place in the abstract in my view.

Great point - we have added a line to the abstract to highlight this point. "These results have implications for the thermal forcing (and thereby mass balance) of the Antarctic Ice Sheet."

Line 411: Relevant to explore for process understanding, but I don't think 'analog' is the right word here given the different orbital parameters, and CO2 levels and also the point above that AMOC collapse has been predicted by some studies this century.

Good point. To clarify, we do not mean the entire global climate of 127ka is analogous to future climate, but just the thermal forcing near the ice sheet in 127ka (slight warming) is more analogous than the 127kaFW cooling due to freshwater forcing. We've changed the language here to make this clear: "Based on these simulations, we suggest that the subsurface ocean response near the AIS in the 127ka run provides a more relevant analog to future climate than changes simulated by either the 127kaFW run or other simulations that include large freshwater fluxes in the North Atlantic."

Line 415: model*s*, the only typo I found in the whole manuscript!

Nice catch. Fixed.

References

Auger et al., Nature Commun., 2021, https://www.nature.com/articles/s41467-020-20781-1

Brown and Galbraith, Clim. Past, 2016: https://doi:10.5194/cp-12-1663-2016).

Ditlevsen & Ditlevsen, Nature Clim. Change, 2023: https://doi.org/10.1038/s41467-023-39810-w

Herraiz-Borreguero & Naviera Garabato, Nature Clim. Change, 2022 (DOI:10.1038/s41558-022-01424-3)

---

## Author Response (AR2)

Dear Mira Berdahl and coauthors

Thank you very much for providing a revised version for your manuscript. The reviewers are very satisfied with how you addressed their comments and with the new version of the manuscript. Nevertheless, one of them has pointed out a few minor technical issues and I have noticed a few more which I point out below. I am therefore accepting your manuscript for publication subject to technical corrections.

I thank all three reviewers for their effort and constructive comments.

Best regards
Marisa

Comments refer to the track-changes manuscript:

line 12: "milliennia" should be "millennia"
line 34: "(estimates suggesting it occurred between 135 ka and 130 ka (Clark et al., 2020))" - please try to rephrase avoiding the double parenthesis
line 60:" (0.2 Sv (sea level rise ~ 17 mm/year))" - same as above, please try to rephrase avoiding the double parenthesis
line 103: "three to eight thousand" should be written with numbers; also correct "deglacaton"
line 125: "temps" should be "temperatures"; sorry I didn't notice this before
Table 2 caption and below: "statistically significant at the 95th confidence interval" should be "statistically significant at a 5% significance level"
line 147: Two Sverdrups should be 2 Sv
line 182: SST's → SSTs
Caption of Figure 3: South should be south
line 368: Pauling et al (2016) and Pauling et al (2017) should appear in parenthesis

We thank the reviewers and the editor for their careful edits and thoughtful consideration of this manuscript. All of the above comments have been addressed now.